# Four-dimensional aircraft emission inventory dataset of Landing and takeoff cycle in China (2019–2023)

Jianlei Lang[1,2], Zekang Yang[1], Ying Zhou[1], Chaoyu Wen[1], Xiaoqing Cheng[1]

[1]Key Laboratory of Beijing on Regional Air Pollution Control, College of Environmental Science and Engineering, Beijing University of Technology, Beijing, 100124, China

[2]Beijing Laboratory for Intelligent Environmental Protection, Beijing University of Technology, Beijing, 100124, China

*Correspondence to*: Ying Zhou (y.zhou@bjut.edu.cn)

**Abstract**

The rapid growth of the aviation industry has resulted in aircraft emissions during landing and takeoff (LTO), which have direct and increasingly adverse impacts on air quality and human health. An accurate and high-resolution LTO emission inventory is crucial for investigating these adverse effects, with the LTO emission having unique three-dimensional spatial characteristics and typical hourly temporal variations. This study integrated the emission calculation and flight trajectory recognition methods to establish a four-dimensional aircraft emission inventory dataset of China's LTO cycle (4D-LTO emission inventory dataset) from 2019 to 2023. The dataset has a high spatial-temporal resolution (hourly, $0.03° \times 0.03° \times$ 34 height layers) and incorporates calculation emissions accurately. Moreover, the actual taxi out/in time for each flight was determined by a statistical model of taxi time and some aircraft in schedule based on 38,000,000 flights. Each flight's climb/approach time was also obtained based on mixing layer height (MLH) and the height-time nonlinear relationship. Additionally, we calculated the LTO emission for China's flight, establishing the hourly emission inventory based on each mode's running time, emission index, and fuel flow. We obtained the flight trajectory core of each airport based on measured flight trajectories and the Density-Based Spatial Clustering of Applications with Noise (DBSCAN) to depict the spatial distribution. Then, each flight's takeoff/landing direction and trajectory were identified from the wind direction and relative departure/arrival airport position. The findings indicate that the impact of COVID-19 has reduced the LTO number in 2020–2022 to 73.1%, 77.6%, and 48.7% of 2019 levels, respectively. However, by 2023, the LTO number has rapidly bounced back to 95.3% of 2019 levels. The recovery rate during daytime (6:00–23:00) was 41.6% higher than night-time (0:00–5:00). The emissions of various pollutants were measured as follows: HC, CO, NOx, PM, and $SO_2$ are 3.2 Gg, 46.1 Gg, 62.3 Gg, 1.1 Gg and 18.4 Gg. LTO emissions' horizontal characteristic is the distance along the runway and spread. This elongated distribution will be hidden if a rough grid (e.g., $0.36°\times0.36°$) and the emissions are evenly distributed. Moreover, LTO emissions height characteristic 'decreases with height,' and the maximum height varies with MLH. Emissions above the standard height set by the International Civil Aviation Organization standard height (~915 m) are not estimated. For example, NOx emissions above 915 m during various months make up an average of 24.6% (9.9%–37.5%) in the LTO cycle, indicating the emissions are significantly underestimated when using the ICAO method. Compared with conventional spatial

allocation methods, our dataset provides a more accurate representation of the actual LTO situation in both horizontal and height at different times. Our 4D-LTO emission inventory dataset and its adaptable methodology are valuable resources for researching temporal and spatial variations, air quality, and health impacts of aircraft emissions in the LTO cycle. The dataset can be accessed from https://doi.org/10.5281/zenodo.13908440 (Lang et al., 2024).

## 1 Introduction

The aviation industry has experienced rapid growth in recent years. However, aircraft emit pollutants such as $NOx$, $CO$, $SO_2$, $HC$, and PM during operation, affect air quality, and have adverse effects on human health and human life (Wang et al., 2022; Dissanayaka et al., 2023; Pandey et al., 2024). It has been estimated that 8000–58000 premature mortalities each year are attributable to aviation emissions (Barrett et al., 2010; Eastham and Barrett, 2016; Quadros et al., 2020; Eastham et al., 2024). Establishing an accurate aircraft pollutant emission inventory is crucial to investigate the impact of aircraft emissions on the environment and health.

According to the standard height (~915 m) of the mixed layer height (MLH), the International Civil Aviation Organization (ICAO) divides the flight process of the aircraft into the Landing and Takeoff (LTO) cycle phase and Climb-Cruise-Descend phase (Kurniawan and Khardi, 2011; Bao et al., 2024). The LTO cycle occurs near the ground and affects the air quality near the airport and the health of the surrounding residents (Christodoulakis et al., 2022). Therefore, many studies (Kurniawan and Khardi, 2011; Zhou et al., 2019; Cui et al., 2022) have focused on aircraft emissions during the LTO cycle. Unlike road, rail, and sea transportation, the flight process in the LTO cycle has prominent four-dimensional (4D) characteristics. For example, aircraft emissions have typical hourly temporal variations due to the impact of human activities. Moreover, the aircraft's unique three-dimensional (3D) flight trajectory (Koudis et al., 2017) makes it a distinctive 3D linear emission source. As a result, comprehensive spatial and temporal consideration is crucial for accurately calculating the pollutant emissions of aircraft in the LTO cycle.

For calculating pollutant emission of aircraft in the LTO cycle, most of the current research is based on the ICAO standard method (Kurniawan and Khardi, 2011; Cui et al., 2022). ICAO stipulates that the LTO cycle is divided into four modes: take off, climb, approach, and taxi, reflecting that the standard operation time of each mode is 0.7 min, 2.2 min, 4 min, and 26 min, respectively (ICAO, 2011). However, unchanged running time is inconsistent with the actual aircraft operation process (Xu et al., 2020) because the running time of different modes in the LTO cycle is influenced by runway congestion (Badrinath et al., 2020) and MLH variations (Peace et al., 2006; Nahlik et al., 2016). Therefore, relying on the ICAO method may lead to high uncertainties. An alternative approach is to use accurate flight data, such as ADS-B data (Klenner et al., 2022; Zhang et al., 2022), which can significantly improve the accuracy of pollutant emission calculations. However, this method still has problems, such as difficulty obtaining actual aircraft data and limited application range. Therefore, multi-year, hourly aircraft emission datasets that accurately reflect reality are still lacking.

In the air quality simulation, addressing the issue of pollutant emission inventory in the LTO cycle of aircraft in spatial is a significant challenge. Previous studies have primarily focused on the environmental impact of pollutant emissions from aircraft during the LTO cycle (Yim et al., 2015; Yang et al., 2018; Bo et al., 2019). However, most of these studies have allocated these emissions to the grid where the airport is without considering the altitude, longitude, and latitude of the emissions location. While this allocation method is suitable in rough grid settings, using a finer grid to reflect aircraft emissions' environmental impact more accurately leads to more significant errors (Kumar et al., 1994; Arunachalam et al., 2011; Woody et al., 2013). Therefore, considering the actual flight characteristics of aircraft is vital to obtain more realistic spatial characteristics of aircraft pollutant emissions and improve the accuracy of air quality simulation. The impact of aircraft emission heights and horizontal position distribution modes on air quality varies widely, as demonstrated by various studies (Unal et al., 2005; Wolfe et al., 2016; Woody et al., 2016; Lawal et al., 2022). Zhang et al. (2024) conducted air quality simulations based on actual flight trajectories in the ADS-B data for typical regions. However, this method is limited by the availability of flights with ADS-B data and cannot be widely applied. Consequently, there is still a lack of aircraft 4D emission inventory datasets in the LTO cycle that accurately reflect actual 3D flight trajectories and their dynamic nature over time.

As the world's second-largest aviation market (CAAC), China contributes 13% of global flight operations (Graver et al., 2020), and accounting for 7.8% to 23.5% of global aviation-related pollutant and carbon emissions (Ma et al., 2024; Teoh et al., 2024). Improving the accuracy of aviation emission estimates and enhancing temporal-spatial resolution in China can not only promote the green development of the Chinese aviation industry but also exert a far-reaching impact of global aircraft pollution mitigation. The period 2019–2023 is a unique period of COVID-19 outbreak. Therefore, we have developed a 4D aircraft emission inventory (4D-LTO emission inventory dataset) for mainland China's takeoff and landing (LTO) cycle from 2019 to 2023. This inventory provides detailed and accurate emissions calculations and flight trajectory recognition. It offers high spatial and temporal resolution, with a horizontal resolution of 0.03° × 0.03° and 34 layers of height resolution from 0 m to 15,668 m. Our dataset and methodology are valuable resources for studying the temporal and spatial variations, air quality, and health impacts of aircraft emissions during the LTO cycle.

## 2 Methodology and data

Figure 1 illustrates the process of establishing the 4D-LTO emission inventory dataset, including the methods, the primary dataset, and the final output information. We developed the 4D-LTO emission inventory dataset in four steps:

(1) Accurately estimate the pollutant emissions of aircraft in the LTO cycle;

(2) Identifying the 3D flight trajectory of aircraft in the LTO cycle;

(3) Spatial and temporal allocation method of the 4D-LTO emission inventory dataset;

(4) Compared with the conventional spatial allocation method.

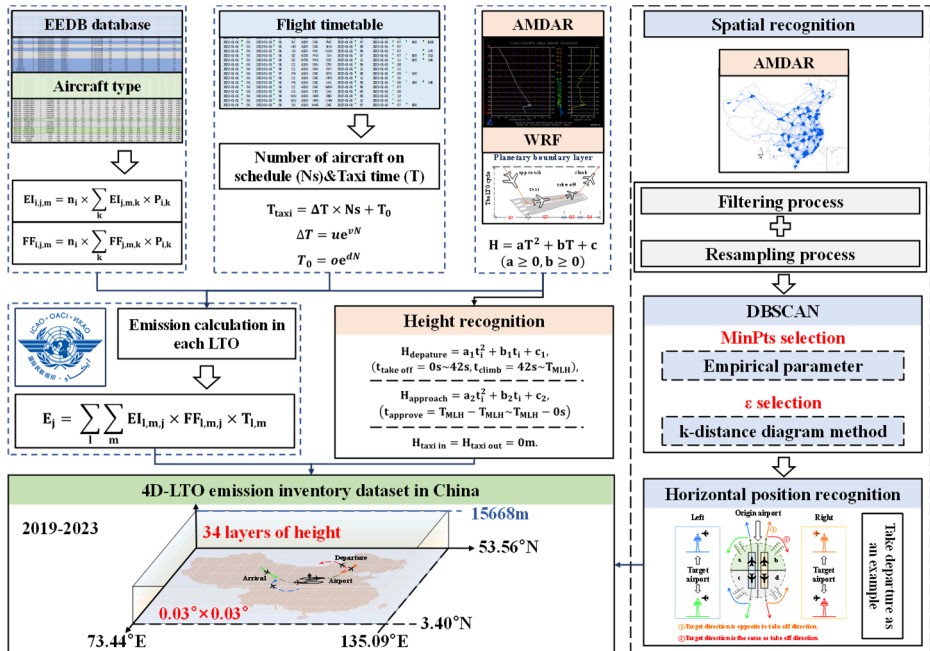

**Figure 1 Schematic of the method used to develop a high spatial and temporal resolution aircraft 4D emission inventory in the LTO cycle. It includes a detailed pollutant emission calculation method, flight trajectory recognition method (The horizontal recognition method takes the departure process as an example.), and LTO cycle emission inventory allocation method.**

The 4D-LTO emission inventory dataset is a grid emission inventory dataset established by combining the aircraft emission calculation method and the flight trajectory identification method for the LTO cycle (described in Sections 2.1 and 2.2.). The spatial-temporal allocation method is introduced in Section 2.3, and the comparison method is described in Section 2.4.

## 2.1 Aircraft LTO cycle emission calculation

The emission index, fuel flow, and running time of different flight modes are used to estimate the civil aircraft emissions in China based on the ICAO method (Kurniawan and Khardi, 2011; Bao et al., 2024). The calculation method is as in (1):

$$E_j = \sum_l \sum_m EI_{l,m,j} \times FF_{l,m} \times T_{l,m}, \tag{1}$$

where $E_j$ is the emission (g) of pollutant j (including NOx, HC, SO$_2$, CO, and PM); $EI_{l,m}$ is the emission index (g/kg) under m mode (take off, climb, approach, and taxi) of LTO $l$. $FF_{l,m,j}$ is the fuel flow (kg/s) of pollutant j under m mode of LTO $l$, and $T_{l,m}$ is the running time under the m mode of LTO $l$.

The actual parameters of each flight should be used to calculate the emissions. However, complete data cannot be obtained due to problems such as incomplete data recording and recording errors. As a result, we have used different methods to approach the emission index, fuel flow (Section 2.1.1), and running time (Section 2.1.2 and 2.1.3) in (1) for the actual situation and estimate more accurate emissions.

### 2.1.1 Aircraft-engine matching

The aircraft's emission factor and fuel flow depend on its engine type, and the same aircraft type can be equipped with different types of engines. Thus, we collected as much detailed engine configuration information as possible for various aircraft types to improve the accuracy of the calculations. The matching method is divided into three steps:

1) We counted all aircraft types departing from or arriving in China from 2019 to 2023 using the flight information dataset from the VariFlight, querying the aircraft type corresponding to each aircraft code.

2) We carefully counted the China airlines, civil aviation fleet in service information, points type statistical engine number, type, and proportion through a comprehensive search in flights associate dynamic query (VariFlight) and Civil Aviation Leisure Station (CALS).

3) We weighed the EI and FF of each aircraft type to obtain the value (Yang et al., 2018) using the information of all aircraft types and the proportion information of different engine types for each aircraft type, combined with the emission index (EI) and fuel flow (FF) data of each engine type given in the ICAO Aircraft Engine Emissions Databank (EEDB).

The EI of an aircraft type in different modes were calculated as (2):

$$EI_{i,m,j} = n_i \times \sum_k EI_{k,m,j} \times P_{i,k}, \tag{2}$$

where $EI_{i,m,j}$ is the emission index of aircraft type $i$ in mode $m$ (g/kg) of pollutant $j$ (NOx, HC, and CO); $n_i$ is the number of engines fitted to aircraft type $i$; $EI_{k,m,j}$ is the emission index of engine $k$ in mode $m$ of pollutant $j$ (g/kg); and $P_{i,k}$ is the proportion of aircraft type $i$ equipped with engine $k$.

The FF of an aircraft type in different modes were estimated as (3).

$$FF_{i,m} = n_i \times \sum_k FF_{k,m} \times P_{i,k}, \tag{3}$$

where $FF_{i,m}$ is the fuel flow of aircraft type $i$ in mode $m$ (kg/s); $FF_{k,m}$ is the fuel flow of engine $k$ in mode $m$ (kg/s); and the definitions of other parameters are similar to those used in (3).

In addition, the first-order approximation 3.0 (FOA3.0) (Wayson et al., 2009) method was used to recalculate the EI of PM, which is not included in EEDB. The emission factor of $SO_2$ is related to the sulphur content of jet fuel, so we used 3.868 g/kg as the emission factor of $SO_2$ (GB6537). In summary, the reference of the EI for different pollutants were shown in Table S1.

### 2.1.2 Climb and approach time calculation

The daily maximum mixing layer height (MLH) serves as a key parameter for determining climb and approach modes of flight operations, and varies with region and time. Given data accessibility constraints, we substituted daily maximum MLH with the daily maximum planetary boundary layer height (PBLH), which shares analogous dynamic characteristics. The three steps for calculating climb and approach times are as follows.

1) Different airport daily maximum PBLHs in 2019–2023 were obtained based on Weather Research and Forecasting (WRF) model. The model parameter settings are described in our previous study (Wen et al., 2023).

2) The relationship between flight time and height were established. In our previous study (Zhou et al., 2019), the relationship for different airports in different months under the approach and climb mode was built based on Aircraft Meteorological Data Relay (AMDAR) data. AMDAR includes the aircraft's position (longitude, latitude, and altitude), speed, and associated meteorological parameters which were collected by the aircraft navigation system. The recording intervals are set at 6 s for the first 60 s of the climb phase, followed by once every 35 s thereafter, and once every 60 s during the descent phase. The form of the relationship for climb and approach mode can be found in Text S1 of SI. The $R^2$ ($p < 0.001$) of the functional relationships of the climb and approach mode were above 0.93.

3) Each flight's actual climb and approach times from 2019 to 2023 were calculated based on the relationship of climb and approach mode mentioned above, and the daily maximum PBLH at different airport.

### 2.1.3 Taxi in and taxi out time calculation

ICAO specifies the taxi mode's running time (taxi out 19 min; taxi in 7 min). However, the actual taxi time varies based on airport flight schedules during actual operation, and using a fixed time can lead to emissions calculation uncertainty. Therefore, the actual taxi time data was used to calculate the aircraft's taxi emissions accurately. The actual taxi time data were obtained from the VariFlight based on the information of ADS-B system, which is recognized by researchers as a reliable data source (Klenner et al., 2022; Zhang et al., 2022; Teoh et al., 2024).

Since not all aircraft record the actual taxi time and the actual taxi time is not publicly available (Table S2), this study has collected all available taxi time data, with coverage rates ranging from 47.9% to 67.0% during 2019–2023, which were summarized in Table S2. It can represent the taxiing conditions of aircraft at different airports with different operating scales. The missing taxi time were supplemented based on the hourly-airport difference relationship model between taxi time and aircraft number of schedules. The functional relationship between the number of aircraft on schedule and the taxi time is as follows:

$$T_{taxi} = \Delta T \times Ns + T_0, \tag{4}$$
$$\Delta T = u \cdot e^{vN}, \tag{5}$$
$$T_0 = o \cdot e^{dN}, \tag{6}$$

where, $T_{taxi}$ is the taxi out (in) time of each flight (s); $\Delta T$ is an increase in taxi time per $Ns$ (s/aircraft). $T_0$ is the initial taxi time (s), and $Ns$ is the number of aircraft on schedule in an hour. $N$ is the annual average aircraft departures/arrivals number for each hour; $u$, $v$, $o$, and $d$ are the airport-specific constant.

Lang et al. (2024, in review) reported the hourly airport difference relationship model between $T_{taxi}$ and $Ns$ at different airports in 2019. The taxi time relationship construction method was used to update the database from 2020 to 2023. The performance of the taxi time calculating model for different airports were shown in Figure S1 and Table S3, taking 2023 as an example. In addition, in this study, Beijing Capital International Airport (PEK) is chosen as a case to test the performance

of taxi time model (Figure 2(a) and Figure 2(b)) under diverse flight situation (e.g., high-density scenarios), due to the centralized terminal layout and relatively frequent ground congestion (Liu et al., 2024). Taking 12:00 from 2019 to 2023 for the Beijing Capital International Airport (PEK) as an example, Fig. 2(a), Fig. 2(b) represent the comparative verification of
function relationships for taxi in and taxi out in different years. We observed a strong correlation between taxi time and the number of scheduled aircraft, regardless of whether it is taxi in or taxi out. The significance level ($p < 0.001$) indicates a strong relationship. The $R^2$ for taxi out ranges from 0.87 to 0.98, and for the taxi in mode, it ranges from 0.96 to 0.99. The model has a good effect on taxi in or out mode at different years, indicating that the model reflects the real taxi time variation. If the relationship between taxiing time and the number of aircraft scheduled cannot be fitted at a certain time due to lack of
records, Table S4 presents the exponential relationship of $\Delta T$ and $T_0$ at different years. In addition, Fig. 2 also provided the five-year exponential relationship of $\Delta T$ and $T_0$, which could be a reference for the other study with no fitting data. We also calculated the coefficient of variation (CV, 30.4% for taxi out and 10.4% for taxi in operations) between the $\Delta T$ and $T_0$ estimation result from five-year model and specific year model, at a representative flight number of 20 (common across all study years). Compared with the actual taxi time, the estimation error of the model result (11.4% for taxi in mode and 20.4%
for taxi out mode) is lower than the result based on the fixed ICAO standard taxi time (27.8% for taxi in mode and 22.0% for taxi out mode). $\Delta T$ and $T_0$ estimation models only are used in the situation that the $\Delta T$ and $T_0$ could not be counted due to the lack of the record.

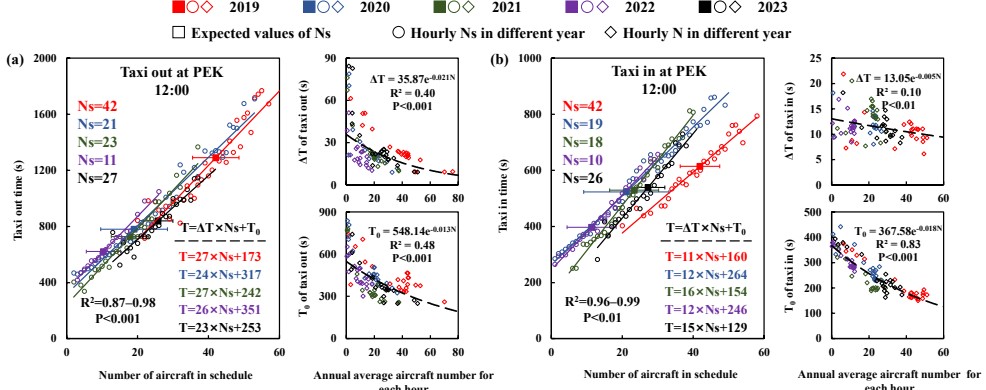

**Figure 2: The linear function relationship between taxi time (T) and the number of aircraft in schedule at each hour (Ns), the**
**exponential function relationship between ΔT and T0, and the annual average departure aircraft number for each hour (N). (a) Taxi out. (b) Taxi in.**

## 2.2 Aircraft emissions 3D trajectory identification

This study was divided into two steps to identify the 3D spatial location of aircraft emissions in China during 2019–2023: (1) The flight altitude identification (Section 2.1.1), (2) Each flight's horizontal trajectory identification (Section 2.1.2).

### 2.2.1 Flight altitude identification

The relationship between flight height and time (Zhou et al., 2019), which has been introduced in Section 2.1.2, was used to identify the altitude at different times for each LTO cycle. The daily maximum PBLH was used to identify the maximum height of each LTO cycle at different airports. When the taxi in and out is 0 m, the takeoff is from 0 m to 152 m (ICAO). The climb is from 152 m to PBLH, and the approach is from PBLH to 0 m. By integrating the altitude information with the emission inventory data established in Section 2.1, we were able to further vertically stratify the pollutant emission inventory during the LTO cycle.

### 2.2.2 Flight horizontal trajectory identification

We established the flight trajectory database of each airport in China based on the Density-Based Spatial Clustering of Applications with Noise (DBSCAN). Moreover, each flight's trajectory was identified based on the relative position of the departure airport, the arrival airport, and the wind direction. The clustering method is used to screen out flight trajectories with similar characteristics from many actual flight data. This approach helps determine the grid location of aircraft emissions (Gariel et al., 2011; Bombelli et al., 2017). DBSCAN is a density-based clustering algorithm widely used in machine learning and data mining (Chen et al., 2021; Tekin et al., 2024). For the transportation industry, it is used for the identification research of road traffic, ship, and aircraft trajectories (Gui et al., 2021; Deng et al., 2023; Li et al., 2023). The DBSCAN algorithm belongs to unsupervised learning, and the initial value setting does not significantly affect the clustering results (Ventorim et al., 2021). As a result, the DBSCAN algorithm is well suited for flight trajectory clustering processing with unclear information, such as the number of clusters and distribution characteristics (Murça et al., 2018; Giovanni et al., 2024).

Before clustering, flight trajectory data belonging to the LTO cycle should be extracted from a vast amount of information in AMDAR. First, the climb and approach modes in the LTO cycle are screened according to the ascending and descending symbols in AMDAR information. Second, each flight trajectory was divided into airport ownership according to the airport's location. Finally, the horizontal position information (time, longitude and latitude) of each flight trajectory in the climb and approach modes of different airports was obtained as the input information for flight trajectory clustering.

The DBSCAN algorithm relies on two input parameters, the minimum number of samples (MinPts) and distance threshold (ε), to cluster the data space based on three basic concepts: directly density-reachable, density-reachable, and density-connected (Sander et al., 1998). MinPts determines the minimum number of points required to form a dense region, while ε specifies the maximum distance between two points to be considered as within the same neighbourhood.

DBSCAN is good at calculating the distance between points, but it is difficult for DBSCAN to process the flight trajectory with time attribute in this study (Chen et al., 2021). Therefore, we use the Euclidean norm to compute the distance between the two sets of flight trajectories. The premise of using the Euclidean norm is to keep the time interval of each set of flight trajectories the same. However, the time interval of each flight trajectory sequence is not the same because of each flight's

trajectory difference and recording delay. As a result, we conducted unified processing of each departure and arrival trajectory through the resampling method. Too low and too high of sampling points will make the location feature information unclear and increase the computational complexity of clustering processing, respectively. Based on all actual flight data from 2019–2023, during the LTO cycle, departure was within 480 s and arrival was within 1200 s. To comprehensively considered the recording intervals of AMDAR data, the uniformity across departure and arrival phases, and computational complexity, we set the sampling points of each trajectory to 25.

MinPts selection: Since the average number of sample trajectories varies in different airports, the MinPts must be taken separately for various airports. For each airport, the MinPts are respectively taken in the range of 6 to 10, and the clustering effect is observed, from which the appropriate MinPts are selected. ε selection: The method of k-distance (Garg et al., 2020) graph was used to select the appropriate ε. The k-distance curve first calculates the distance between each trajectory in the data and the trajectory with the nearest k, then arranges the k-distances of all trajectories in descending order and draws the curve. Moreover, k values for different airports are the same as MinPts, and the ε values are based on the apparent inflection point in the k-distance curve.

Figure S2 shows the overall performance of the trajectory clustering model for different airports. In addition, in this study, Shanghai Pudong International Airport (PVG) is chosen as a case to test the performance of trajectory cluster (Figure 3) under the normal flight trajectories scenario, as well as the deviations in flight trajectories due to the crosswinds and typhoons which is the challenge for the robustness of the trajectory cluster algorithm (Wang et al., 2017; Xu et al., 2020). This study evaluated the core flight trajectory of departure and arrival since the flight trajectory is a series of latitude and longitude information with time series characteristics. This approach uses the DBSCAN clustering method by splitting the flight trajectory of departure and arrival into two directions, latitude, and longitude, considering the three indexes of R and MAE. In longitude, the correlation between identified core trajectories and actual trajectories is more significant than 0.80 (departure: 0.865–0.992; arrival: 0.811–0.997). In latitude, the correlation between identified core trajectories and actual trajectories is more significant than 0.94 (departure: 0.947–0.995; arrival: 0.941–0.992). The identified core trajectory is consistent with the actual flight trajectory, indicating that the core trajectory can reflect the actual flight situation. In longitude, the MAE between identified core trajectories and actual trajectories is less than $0.05°$ (departure: $0.01°$–$0.02°$; arrival: $0.02°$–$0.05°$). In latitude, the MAE between identified core trajectories and actual trajectories is less than $0.05°$ (departure: $0.01°$–$0.02°$; arrival: $0.02°$–$0.05°$). Although the clustering results are uncertain, they can still provide vital information for the 3D grid location of aircraft emissions.

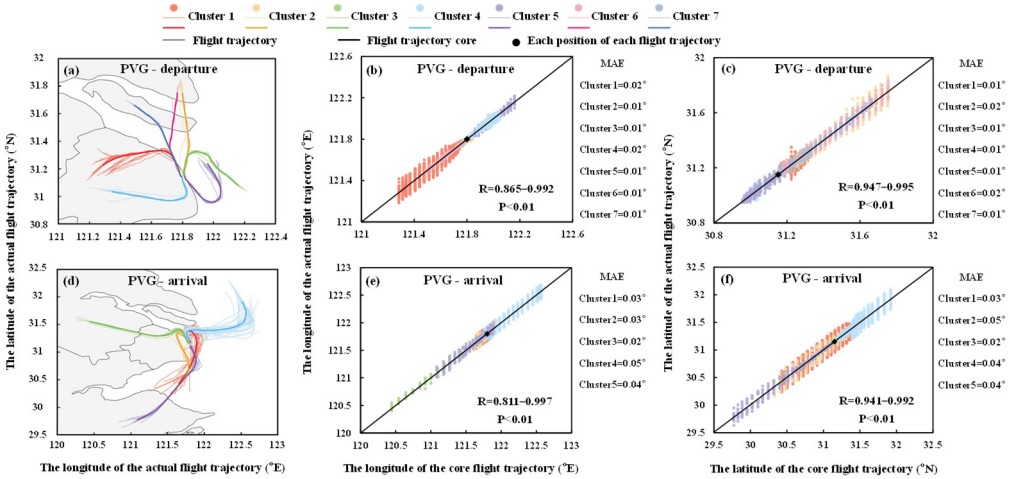

**Figure 3: (a)** Departure flight trajectories of different clusters. **(b)** Consistency in longitude of departure between the core and actual flight trajectories. **(c)** Consistency in latitude of departure between the core and actual flight trajectories. **(d)** Arrival flight trajectories of different clusters. **(e)** Consistency in longitude of arrival between the core and actual flight trajectories. **(f)** Consistency in latitude of arrival between the core and actual flight trajectories.

The airports with multiple runways will assign a suitable runway for each flight based on the relative location of the departure and arrival airport (Yin et al., 2022; Sekine et al., 2023). This decision is made considering the need for aircraft to operate against the wind as per the Chinese Meteorological Administration; CAACNEWS. Therefore, these flight characteristics were combined to identify the horizontal trajectory of each flight in the LTO cycle (Horizontal position recognition in Fig. 1). First, all the flight trajectory clusters corresponding to the departure/ arrival airport are selected from

the flight trajectory database obtained by DBSCAN method. Second, trajectories from the runway are chosen close to the target airport. Third, the aircraft takes off against the wind principle, selecting trajectories on which side of the runway based on the wind direction information at the moment of the departure/arrival airport. Finally, the final trajectory is selected by the target direction being opposite or the same as the takeoff direction.

## 2.3 Temporal and spatial identification of 4D emission inventory

Gridded emission information is often required for air quality and climate simulation models or refined prevention and control of pollutants. Therefore, the obtained aircraft pollutant emission inventory of the LTO cycle in China during 2019–2023 was processed into hourly, 3D grid pollutant emission data with a horizontal resolution of 0.03° × 0.03° and a height resolution of 34 layers from 0 m to 15668 m.

### 2.3.1 Aircraft emission temporal allocation

The 4D-LTO emission inventory dataset has an hourly temporal resolution. According to (1), the emissions of each pollutant in different modes of each LTO cycle are calculated separately, and the emission for each hour of the LTO cycle at different airports is the sum of the pollutant emissions generated by all departure and arrival at that airport during that hour. In

addition, the daily, monthly, and yearly total emissions are the sum of all LTO cycles on that day, month, and year to further analyse the temporal variation of pollutant emission.

## 2.3.2 Aircraft emission spatial allocation

For the horizontal resolution, most airport runways are approximately 3–4 kilometers (CAAC) in length and certain pollutants (such as CO) are predominantly emitted during taxiing, i.e., on the runway. $0.03°×0.03°$ is capable of reflecting the horizontal distribution characteristics of aircraft emission. In addition, $0.03° × 0.03°$ is also a common resolution for air quality models. Therefore, the horizontal resolution of the 4D-LTO emission inventory is $0.03° × 0.03°$ with the latitude and longitude range of 3.40°N–53.56°N and 73.44°E–135.09°E, respectively.

For the altitude resolution, while ICAO defines the LTO cycle with a fixed mixing layer height (915 m), in reality, the mixing layer height varies significantly with region and time, leading to variations in the altitude range of the LTO cycle. Therefore, to better reflect the vertical distribution of aircraft emissions above 915 m during the LTO cycle, this study set the altitude range from 0 m to 15668 m. In addition, to ensure that the emission inventory can be effectively used in air quality models, this study used the air quality model commonly used 35-layer sigma stratification strategy (Wolfe et al., 2016). Therefore, the altitude resolution was divided into 34 layers from 0 m to 15668 m (0.0 m–38.3 m, 38.3 m–76.7 m, 76.7 m–115.3 m, 115.3 m–154 m, 154 m–231.8 m, 231.8 m–310.3 m, 310.3 m–389.3 m, 389.3 m–469 m, 469 m–549.3 m, 549.3 m–630.3 m, 630.3 m–711.9 m, 711.9 m–794.2 m, 794.2 m–960.7 m, 960.7 m–1130.1 m, 1130.1 m–1302.3 m, 1302.3 m–1477.6 m, 1477.6 m–1656.0 m, 1656.0 m–1929.7 m, 1929.7 m–2211.1 m, 2211.1 m–2599.3 m, 2599.3 m–3107.2 m, 3107.2 m–3643.1 m, 3643.1 m–4210.5 m, 4210.5 m–4813.9 m, 4813.9 m–5458.5 m, 5458.5 m–6151.2 m, 6151.2 m–6900.4 m, 6900.4 m–7717.4 m, 7717.4 m–8617.3 m, 8617.3 m–9621.2 m, 9621.2 m–10759.7 m, 10759.7 m–12080.6 m, 12080.6 m–13664.8 m, 13664.8 m–15668 m.).

The 4D-LTO emission inventory dataset was processed by first identifying the emissions information of each flight into a 3D grid through latitude, longitude, and altitude information. Then, the emissions of all flights were summarized within the same hour.

## 2.4 Comparison of our dataset with the previous dataset

Our dataset was compared with the spatial allocation methods commonly used in previous studies. (1) Other studies typically assign aircraft emissions in the LTO cycle according to the standard altitude for each mode as defined by ICAO (Mokalled et al., 2018; Bo et al., 2019 Wang et al., 2023; Zhang et al., 2023). (2) The conventional horizontal distribution method for aircraft emissions in the LTO cycle assumes that aircraft emissions are radially distributed (Lawal et al., 2022). The Federal Aviation Administration (FAA)-recommended the standard climb rate of 200 ft per nautical mile. Therefore, the standard climb rate and ICAO standard altitude determine the horizontal distribution of aircraft emissions around the airport. The running time, altitude, and horizontal range of each mode defined by ICAO are shown in Table 1.

**Table 1: The running time and altitude range of each mode defined by ICAO**

| Mode | Running time (s) | Altitude range (m) | Distance to Airport (km) |
|---|---|---|---|
| Take off | 42 | 0–152 | 0–5 |
| Climb | 132 | 152–915 | 5–28 |
| Approach | 240 | 0–915 | 0–28 |
| Taxi in | 420 | 0 | – |
| Taxi out | 1140 | 0 | – |

## 2.5 Uncertainty calculation

The uncertainty of the 4D-LTO emission inventory dataset is mainly divided into emission calculation uncertainty and spatial location identification uncertainty. This study assumes that the uncertainty of all input parameters follows a normal distribution.

When calculating the emissions uncertainty, this study comprehensively considered the uncertainty of EI, FF and T. The EI and FF are weighted based on the engines data from the EEDB and the engine proportion data for different aircraft types. Therefore, the standard deviation of EI or FF were calculated using the formula (7):

$$\sigma = \sqrt{\sum_k (x_k - \bar{x})^2 \times P_k}, \qquad (7)$$

where $\sigma$ represents the standard deviation of EI or FF, $k$ represents the engine type, $x_k$ represents the EI or FF value of the engine k, $\bar{x}$ represents the weighted average of EF or FF, and $P_k$ represents the proportion of engine k.

The climb and approach time is obtained using the relationship between flight time and flight height (Zhou et al.,2019). Therefore, the standard deviation of the climb and approach time is the combine of the standard deviation of a function fitting parameter a, b, and c. The taxi in/out time is calculated using the formula (4). Therefore, the standard deviation of the taxi in and out time is the combine of the standard deviation of a function fitting parameter $\Delta T$ and $T_0$. This study used the Monte Carlo sampling method to obtain the 95% prediction interval of the emission for different pollutants with 20,000 samples.

The spatial uncertainty during the LTO cycle include the uncertainty of horizontal and altitude position. The standard deviation of the horizontal position is calculated by the error distribution between the flight trajectory clustering result and the actual flight trajectory. The standard deviation of the altitude position is the combine of the standard deviation of a function fitting parameter a, b, and c. This study employed the Monte Carlo method to quantitatively assess the uncertainty of spatial location identification for each hour, with uncertainty ranges derived from 20,000 Monte Carlo simulations at a 95% prediction interval.

# 3 Results and discussion

## 3.1 Total aircraft emissions in the LTO cycle

In 2023, the total emissions of five types of pollutants in the LTO cycle of aircraft in China are as follows: HC is 3.2 Gg; CO is 46.1 Gg; NOx is 62.3 Gg; PM is 1.1 Gg; $SO_2$ is 18.4 Gg as shown in Fig. 4(a). The annual emission of different pollutants in 2023 was 82.9% (HC)–94.1% (NOx) in 2019. However, before 2022 (the last year impacted by COVID-19 and the most affected year), emissions of various pollutants averaged 34.7%–42.8% of 2019. At the end of COVID-19, the 2023 recovery in aircraft emissions shows that the pandemic has not had an irreversible impact on aircraft activities and that emissions from aircraft activity will continue to grow (Teoh et al., 2024). Emissions of pollutants from aircraft, such as NOx and $PM_{2.5}$, are known to cause respiratory and cardiovascular issues (Boningari et al., 2016; Hu et al., 2022; Hou et al., 2024). Therefore, it is essential to pay attention to the growing trend of aircraft activities in order to anticipate and address its potential health impacts.

Figure 4(a), the main emission contribution of HC and CO came from taxi mode (HC: 94.6%; CO: 91.5%) because HC and CO are mainly produced by incomplete fuel combustion, taking 2023 as an example. A large amount of HC and CO are created because the engine's thrust in taxi mode is minimal and the operation time is long (EPA, 1981). The climb is the main NOx emission stage (42.1%). The takeoff with the shortest running time contributes to the second largest NOx emission (25.7%). The taxi with the longest running time contributes the NOx most minor emission of (12.4%), indicating that the emission factor of NOx is highly correlated with the aircraft engine's thrust (Stettler et al., 2011). Although the engine runs for a long time, the NOx emission during taxi mode with a slight thrust is still lower than during the takeoff stage, with the engine running for a short time but at nearly full thrust. For PM and $SO_2$, the emission contribution ratio is similar to the running time of each mode, and the taxi mode with the longest running time contributes 33.1% of PM and 35.1% of $SO_2$. The climb mode contributes 28.4% of PM and 26.6% of $SO_2$, the approach mode contributes 25.7% of PM and 26.4% of $SO_2$, while the takeoff mode with the shortest running time contributes 12.7% of PM and 11.9% of $SO_2$. From 2019 to 2023, among various aircraft types, B738, A320, and A321 have been the top three pollutant emissions (Fig. 4(b)). The top three aircraft types contributed 64.1% of NOx emissions in 2019, taking NOx emissions as an example. However, during the COVID-19 period (2020–2022), the contribution of the top three aircraft types reached 70.3%–70.9%. At the end of the pandemic impact in 2023, the contribution of the top three aircraft types reversed to the 2019 level (55.0%). During the COVID-19 pandemic, many aircraft types ceased operation, including F50, E145, and other regional aircraft, A306, A340, and other wide-body aircraft types, increasing the proportion of the first three types. As the impact of COVID-19 gradually diminished, the discontinued models resumed operation, and the emission proportion of the first three models returned to normal.

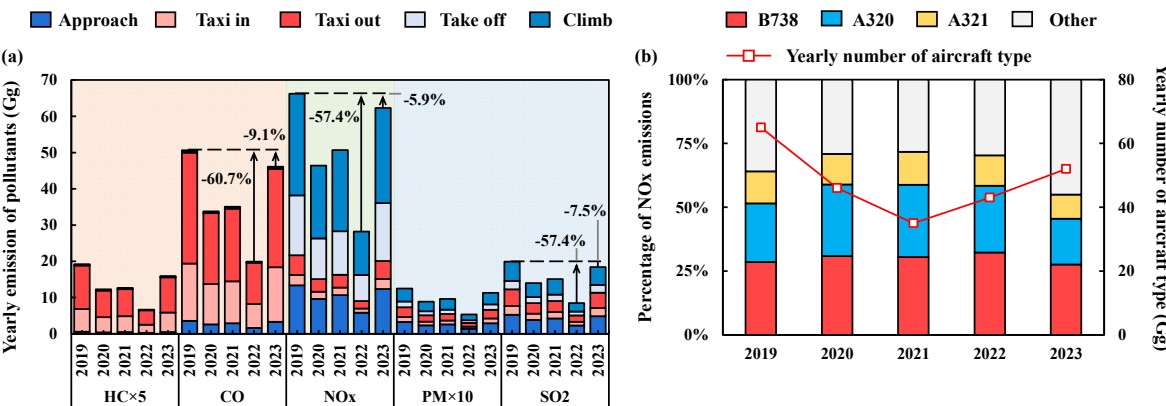

**Figure 4: (a) Total aircraft pollutant emissions of the LTO cycle from 2019 to 2023. (b) Proportion of NOx emissions in different aircraft types from 2019 to 2023.**

## 3.2 Temporal variation of aircraft emissions in the LTO cycle

Figure 5(a) shows the changes in aircraft emissions during the LTO cycle from 2019 to 2023, encompassing the period before, during, and after the COVID-19 pandemic. The baseline year for analysis was 2019, unaffected by COVID-19, and represented regular aircraft activity.

As can be seen from Figure 5(a) and Table S5, from January 20 to February 13, 2020, aircraft activity rapidly dropped to the lowest point owing to the impact of COVID-19, showing that the number of LTO on February 13, 2020, was 84.8% lower than the same period in 2019. In the following months, aircraft activity slowly recovered, returning to the 19-year level in October. As the COVID-19 situation in China entered a recurrent period, from 2021 to the beginning of 2023, the activity of aircraft fluctuated, reflecting five low points (February 12, 2021, August 12, 2021, November 9, 2021, April 4, 2022, November 29, 2022). As the effects of COVID-19 faded from early 2023, aircraft activity gradually returned to 2019 levels. When the impact of COVID-19 is over, abnormal growth is noted in aircraft activity. In May 2021, the number of aircraft LTO increased rapidly compared to the same period in 2019. However, during the same period in 2019, aircraft activity showed a downward trend. From July to October 2023, the number of LTOs exceeded the same period in 2019. This phenomenon occurs because people with unfulfilled travel needs are inclined to engage in revenge tourism following prolonged COVID-19 lockdowns, resulting in increased aircraft activity and a sudden increase in emissions in the short term.

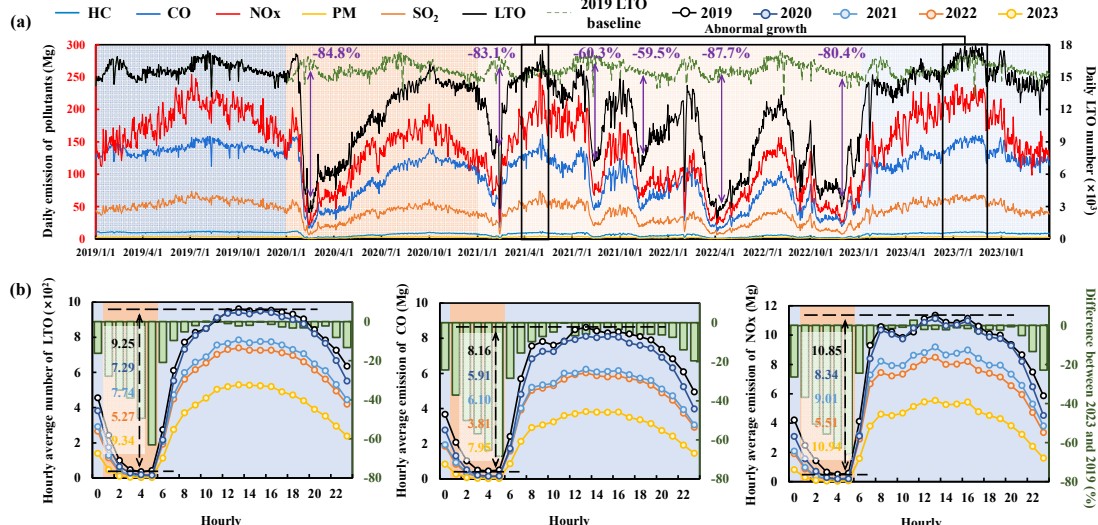

**Figure 5: (a) Daily variation of pollutant emissions and the number of LTO cycles from 2019 to 2023. (b) Annual hourly variation of pollutant emissions and the number of LTO cycles from 2019 to 2023.**

Based on Fig. 5(b), the emission of various pollutants in 2019–2023 varies slightly in hours, with higher daytime (6:00–23:00) and low night-time (0:00–5:00), with the minimum at 4:00 and the maximum at 13:00. The most significant difference of the number of LTO and the emission of pollutants between each hour over the five years occurred at 4:00 and 13:00 in 2022 (251% of LTO, 230% of HC, 244% of CO, 229% of NOx, 249% of PM, 234% of SO$_2$). The difference in pollutant emissions between 2019 and 2023 at each hour shows recovery in 2023. The proportion of LTO numbers that recovered to 2019 levels at night-time was 34.1% lower than during the daytime. Additionally, the recovery rate of the five pollutants emissions at night-time was 39.1%–44.4% lower than at daytime, indicating that aircraft activity resumed significantly better during the day than at night.

**3.3 4D characteristics of aircraft emissions in the LTO cycle**

During the LTO cycle, HC and CO emissions, predominantly emitted during taxi mode (Yang et al., 2018). Consequently, HC and CO emissions are distributed in the first layer of the grid where the runway is located. NOx is an important contributor to overall aircraft emissions and has a significant impact on air quality (Zhang et al., 2024). Furthermore, the spatial distribution of PM and SO$_2$ emissions from aircraft is similar to that of NOx. In summary, this study mainly analyzed the spatial distribution of NOx emissions.

This study calculates hourly aircraft emissions in LTO cycles at various airports in China during 2019–2023 based on the combining emission calculation method and flight trajectory recognition method, establishing a 4D aircraft NOx emission inventory (hourly, 0.03°×0.03°×34) of LTO cycle for in China (Fig. 6 and Fig. 7).

Figure 6(a) signifies the horizontal distribution of yearly NOx emissions in prefecture-level cities and airports during 2019–2023. Compared with 2019, emissions in most regions affected by COVID-19 decreased significantly in 2020–2022.

Notably, aircraft emissions of prefecture-level cities experienced an average reduction of 43.1% in 2022. As the COVID-19 impact ended in 2023, aircraft emissions of prefecture-level cities recovered with an average increase of 5.07%. Although the aircraft emissions of LTO cycle in prefecture-level cities fluctuated from 2019 to 2023, airport emissions in Beijing, Shanghai, Guangzhou, and Chengdu were the top four, accounting for 24.4% (2022)–32.2% (2023) of national emissions.

Between 2019 and 2023, the number of airports in China increased from 237 to 257 at an average annual growth of 5. However, the newly operated airport significantly increased aircraft emissions in a prefecture-level city. For example, due to the operation of TFU airport, aircraft NOx emissions in Chengdu (4.2 Gg) will be 32.3% in 2023, higher than in 2019. In addition, Chengdu's aircraft NOx emissions were 8.4%–14.3% higher than Guangzhou's during 2021–2023, while in 2019, Chengdu's NOx emissions were 21.5% lower than Guangzhou's when TFU airport did not start operations. The newly operated airport can also affect the original airport in a prefecture-level city. Taking airports in Beijing as an example,

PEK airport's annual aircraft NOx emissions (8.1 Gg) were 101%, ranked second in 2019, higher than CAN airport's (4.0 Gg). However, with PKX airport's operation, PEK airport emissions significantly decreased. In 2023, PEK airport's emissions recovered to 54.7% of the original, while the total emissions of Beijing recovered to 81.4%. In addition, emissions from PEK airport in 2023 were only 15.7% higher than those from CAN, indicating that the newly-operated PKX airport has reduced the emission pressure on PEK Airport.

Taking airports in Beijing and surrounding areas in January 2023 as an example, Fig. 6(b) demonstrates the grid horizontal distribution of aircraft NOx emissions in the LTO cycle. The horizontal distribution characteristics of aircraft emissions in the LTO cycle are influenced by the distance along the runway and how they spread, indicating that emissions will be concentrated in the direction of the runway near the airport. With the increase in flight distance, the emissions caused by aircraft will be dispersed. Aircraft emissions during the LTO cycle are widely distributed around the airport, not even represented by a rough grid (e.g., 0.36° × 0.36°). The elongated distribution characteristics of aircraft emissions indicate that evenly allocating emissions around the airport will cause significant uncertainty. Figure 6(c) shows the differences in aircraft emissions at various airports and times between 0:00 and 20:00 on January 3, 2023, at a four-hour interval. This phenomenon indicates that the horizontal distribution characteristics of aircraft emissions vary significantly at different hours and airports. As a result, the refined aircraft emission inventory in the LTO cycle conforms to the time-by-hour spatial distribution characteristics of aircraft, better reflecting the actual situation of aircraft emissions, which is of great significance for accurately assessing aircraft environmental impact in the LTO cycle.

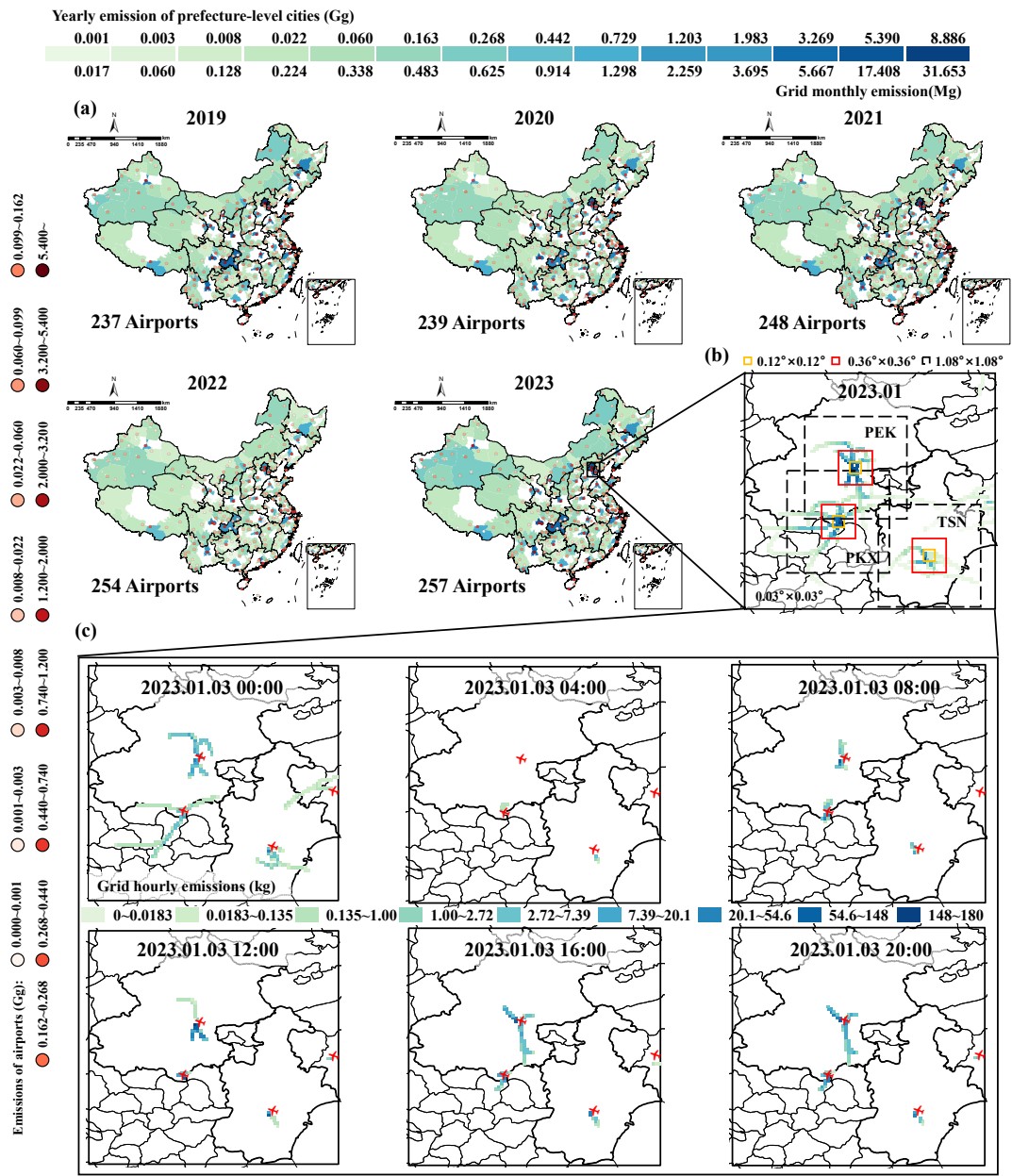

**Figure 6:** (a) Horizontal distribution of yearly NOx emissions in prefecture-level cities and airports during 2019–2023. (b) Horizontal distribution of NOx emissions at airports in Beijing and surrounding areas in January 2023. (c) Horizontal distribution of NOx emissions at airports in Beijing and surrounding areas for different hours in January 2023.

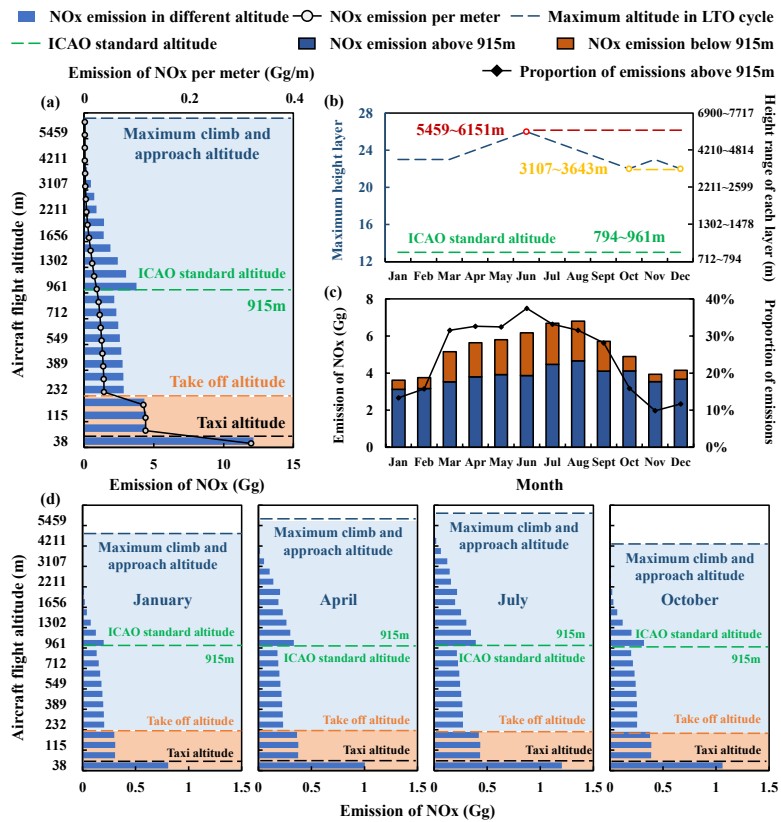

**Figure 7: (a) Height distribution characteristics of NOx emissions in LTO cycle. (b) The maximum altitude layer for different months and the corresponding altitude range. (c) The emissions above and below the height of 915 m in different months and the proportion above 915 m. (d) Height distribution characteristics of NOx emissions from the LTO cycle at different months.**

Figure 7(a) uses the annual NOx emissions in 2023 to demonstrate the height distribution of aircraft emissions in the LTO cycle. In general, the NOx emission of aircraft in the LTO cycle decreases with the increase in altitude. Moreover, the emission per unit altitude significantly decreases between layers 1 and 2 and between layers 4 and 5 due to the different flight altitude ranges in various modes in the LTO cycle. Emissions from layer 1 (0–38 m) include the entire taxi mode as the takeoff mode and approach mode, with the maximum unit height NOx emissions (0.32 Gg/m). Emissions from layers 2 to 4 (38–154 m) include the part of takeoff mode and approach mode, with the unit height NOx emissions of 0.11–0.12 Gg. From layer 5, each layer's NOx emissions (≤ 0.04 Gg/m) include the part of the climb and approach modes. As the emission height increases, the emissions of NOx gradually decrease. The reduction rate gradually increases before layer 14 and decreases after layer 14, indicating that the unit height emissions of each layer above the 14th layer have little difference. In addition, there are significant differences in the height distribution characteristics of emissions in the LTO cycle at different months. Figure 7(b) shows that the maximum emission height in the LTO cycle can reach the 23rd layer (3107–3643 m, November and December)–and the 26th layer (5459–6151 m, June) of 34 layers (0–15668 m). The maximum aircraft emission height in the LTO cycle can reach 4544 m above the ICAO-defined maximum altitude of 915 m due to MLH variation across 12

altitude levels. Figure 7(c) illustrates that the NOx emissions above the ICAO standard height (~915 m) in different months account for an average of 24.6% (9.9%–37.5%) in the LTO cycle. This result indicates that the ICAO method does not account for a significant portion of emissions during the entire LTO cycle. Based on previous study (Köhler et al., 2008; Lee et al., 2013; Yim et al., 2015; Zhang et al., 2023), high-altitude emissions can significantly impact ground-level air quality through atmospheric transport and chemical reactions. When assessing emissions during the LTO cycle and their impact on air quality and health, we must fully consider the contribution of above 915 m emissions. Therefore, using the ICAO fixed flight height will introduce considerable uncertainty when calculating the aircraft emission during the LTO cycle and assessing its environmental impact.

## 3.4 Comparison with the previous allocation method

Figure 8 uses the NOx emissions in January 2023 to show the differences between the 4D-LTO emission inventory and the LTO emission divided in previous study (Mokalled et al., 2018; Bo et al., 2019; Lawal et al., 2022; Wang et al., 2023; Zhang et al., 2023) in terms of height distribution (Fig. 8(a)–(b)) and horizontal distribution (Fig. 8(c)–(e)). Figure 8(a) and Figure 8(b) represent noticeable differences in emissions at different layer heights. Two statistical measures, Mean Absolute Error (MAE) and Mean Absolute Percentage Error (MAPE), were employed to quantify these differences. Two components to the allocation error of the ICAO method are (1) in the range of 154 m–961 m; the ICAO method overestimates the emissions by 63.4%, and the emission difference of different layers is 65.7 Mg–219.3 Mg. The difference increases with a rise in height. (2) With the range of 961 m–4211 m, the ICAO method missed 283.0 Mg of emissions, and the difference decreases with an increase in height (0.0 Mg–125.7 Mg). Figure 8(b) uses PEK, PVG, and CAN to demonstrate the emission height changes of different airports. Different airports' overestimation and missing zones are similar to the height distribution of total NOx emissions. However, the ICAO method misses emissions above 961 m differently for different airports (CAN: 1.9 Mg, PEK: 28.0 Mg, PVG: 5.4 Mg), and the ICAO method overestimates emissions in 154 m–961 m differently in various airports (CAN: 3.4 Mg–11.8 Mg, PEK: 3.4 Mg–10.2 Mg, PVG: 1.5 Mg–10.1 Mg). Compared with the dataset based on the ICAO method, our 4D-LTO emissions inventory dataset can more accurately represent the height distribution of actual aircraft emissions.

In the example of airports in Beijing and surrounding areas, Figs. 8(c) and 8(d) demonstrate that our 4D-LTO emission inventory dataset outperforms the dataset based on the previous radial allocation method, showing an apparent misallocation of emissions. Figure 8(e) quantifies the differences in the horizontal distribution between two emission inventory datasets. Based on the previous radial allocation method, the dataset misallocated 242.7 Mg of emissions in the misallocation zone. Among them, 17.2 Mg of emissions were missing (PEK: 3.0 Mg, PKX: 13.5 Mg, TSN: 0.2 Mg, TVS: 0.5 Mg), and 225.5 Mg of emissions were assigned to the wrong grid (PEK: 122.8 Mg, PKX: 73.7 Mg, TSN: 25.8 Mg, TVS: 3.2 Mg). In the non-misallocation zone, the dataset based on the previous radial allocation method underestimates 41.9% of emissions (PEK: 46.5%, PKX: 37.8%, TSN: 32.9%, TVS: 60.6%). Compared with the dataset based on the previous radial allocation method, our 4D-LTO emissions inventory dataset can better reflect the horizontal distribution of actual aircraft emissions.

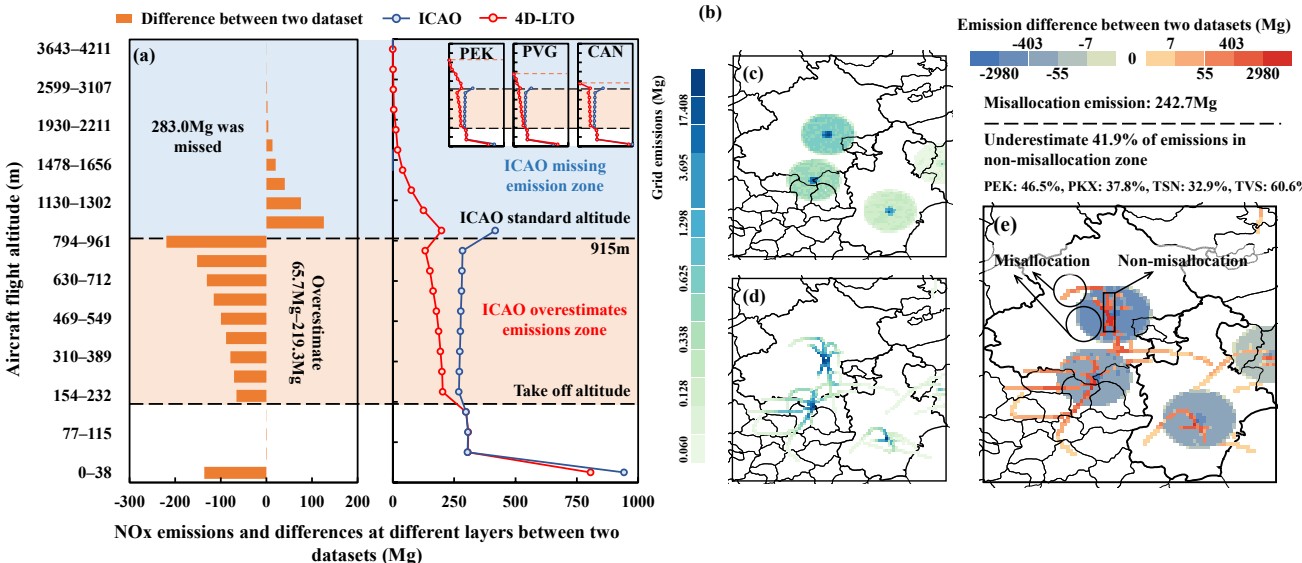

**Figure 8: Comparison of horizontal and height distributions of NOx emissions in January 2023, (a) NOx emissions differences at different heights between two datasets (Mg), (b) NOx emissions distribution at different layers between two datasets (Mg), (c) Distribution of NOx emissions based on previous radial allocation method, (d) Distribution of NOx emissions in 4D-LTO emission inventory dataset, (e) NOx emissions differences at different horizontal grids between two emission inventory datasets.**

**3.5 Uncertainty analysis**

Taking the year 2023 as an example, this study estimated the hourly uncertainty ranges for different emissions in various airports throughout the year, with the average of the uncertainty ranges for NOx, CO, HC, PM, and SO$_2$ being [-17%, 17%], [-16%, 16%], [-6%, 6%], [-8%, 8%], and [-8%, 8%] respectively, and their standard deviations being ($\pm$11%), ($\pm$10%), ($\pm$3%), ($\pm$4%), and ($\pm$4%) respectively.

This study also estimated the hourly average 95% prediction intervals for latitude and longitude of departure and arrival flights at various airports, with the average of [-0.02°, 0.02°] (longitude) and [-0.02°, 0.02°] (latitude) for departure, and [-0.06°, 0.06°] (longitude) and [-0.05°, 0.05°] (latitude) for arrival, respectively, and their standard deviations being ($\pm$0.01°), ($\pm$0.01°), ($\pm$0.01°), and ($\pm$0.01°) respectively. In addition, the result showed that the hourly 95% prediction intervals of climb and approach altitude at different airports within an average of [-78 m, 78 m] and [-134 m, 134 m], respectively, and 505 their standard deviations being ($\pm$49 m) and ($\pm$95 m), respectively.

**3.6 Advantage and limitation**

The previous studies (e.g., Zhang et al., 2022 and Teoh et al., 2024) contributed much in the improvement of the emission estimation by using ADS-B data, laying a solid foundation for further assessing the impact of aircraft emissions on air quality. At percent, the ADS-B data does not fully cover all flights. It's hard to identify the flight trajectories of those flights 510 not covered by ADS-B. Furthermore, using a fixed height range (915 m) for the LTO cycle, introduce errors in the

calculation of pollutant emissions during the LTO cycle. The other emission inventories (e. g., EDGAR, EMEP, AERO2k), are also important sources of aviation emission data. These emissions inventories rely on mainly rely on the flight schedule information, which mainly calculate the total emissions of countries, and does not reflect the detailed four-dimensional emission characteristics.

Based on the emission index and activity level data of each mode during the LTO cycle, we calculate the emissions of each flight by bottom-up method, and give the hourly and three-dimensional spatial distribution of aircraft emissions. It is important information for further assessing aircraft emissions during the LTO cycle and their impact on air quality.

Our 4D-LTO emission inventory dataset reflects the actual spatial and temporal and can be used to accurately assess the air quality impact of aircraft in the LTO cycle, but has several limitations due to data and technical restrictions. (1) According to

our investigation (Airbus; Aircraft Commerce), most aircraft types do not update engine. Some aircraft types (e. g., the aircraft A allow is allowed to be equipped with engine A and B) may change engine configuration proportion during 2019–2023. Given the unavailability of the annual variation of engine configuration for each aircraft type in the existing datasets, this study used the latest proportion data of each aircraft type in different years. (2) The certified engine emission indices derived from the engine manufacturers and reported in the ICAO failed to consider the life expectancy of an aircraft and

meteorological conditions. This may result in errors between the fuel consumption and emissions estimated using these recommend parameters and real-world conditions. Therefore, future research should be conducted on the dynamic emission factors based on the machine age and flight conditions. (3) Our dataset was obtained from a subset of available flight data and near-real flight data based on a model built using real data. This may result in errors between our dataset and real-world conditions. However, this issue will addressed as real-world data becomes more widely available.

**4 Data availability**

The 4D-LTO emission inventory dataset in China from 2019 to 2023 (Lang et al., 2024) presented in this study are freely available at https://doi.org/10.5281/zenodo.13908440.

**5 Conclusions and implication**

This study establishes China's 4D-LTO aircraft emission inventory dataset during 2019–2023 by combining accurate and

generalizable emission methods and flight trajectory identification methods. The actual taxi time is used, and the supplementary value is obtained through the five-year validation (PEK: $R^2 = 0.87$–$0.99$) airport-hourly difference relationship between the taxi in/out time and the number of aircraft schedules. Moreover, the climb and approach time and the attitude of each flight are updated using the MLH and the airport-monthly difference relationship between the flight altitude and time of the climb/approach mode. Finally, the DBSCAN clustering method (PVG: $R = 0.865$–$0.995$ and the

MAE $= 0.01°$–$0.02°$ in departure, $R = 0.811$–$0.997$ and the MAE $= 0.02°$–$0.05°$ in arrival) is used to obtain the flight

trajectory database of each airport based on the massive number of actual flight trajectory data. Then, the flight trajectory of each flight is identified by the wind direction and the relative position of the departure and arrival airport. The data shows that the impact of COVID-19 reduced the LTO number to 73.1% in 2020, 77.6% in 2021, and 48.7% in 2022, compared to 2019. However, in 2023, the emissions of different pollutants quickly bounced back to 82.9%–94.1% of the 2019 levels,

resulting in HC, CO, NOx, PM, and $SO_2$ emissions of 3.2 Gg, 46.1 Gg, 62.3 Gg, 1.1 Gg, and 18.4 Gg, respectively.

Taxi is the most crucial emission stage of HC and CO (94.6% and 91.5% of the emission of the entire LTO cycle), and climb is the primary emission stage of NOx (42.1%). We also find that takeoff with the smallest opera time contributes the second largest emission of NOx (25.7%). Moreover, B738, A320, and A321 are the top three aircraft types that emit pollutants. During the COVID-19 period (2020–2022), the contribution of the top three aircraft types reached more than 70%.

Due to the impact of COVID-19, aircraft emissions in the LTO cycle fluctuate from 2019–2023. After COVID-19 is over, aircraft activity has been abnormal in May 2021 and from July to October 2023. We also find that the number of LTO and pollutant emissions of aircraft slightly differ in hours, exhibiting high rate in the daytime (6:00–23:00) and low rate at night-time (0:00–5:00), with the minimum at 4:00 and the maximum at 13:00. In 2023, the aircraft activity was significantly better during the daytime (95.6% of 2019 in LTO cycle) than at night-time (61.5% of 2019 in LTO cycle).

In the LTO cycle, the horizontal distribution characteristics of aircraft emissions are 'dispersed along the runway,' and the vertical distribution characteristics 'decrease as altitude increases.' We find that aircraft emissions during the LTO cycle are so widely distributed around the airport that even a rough grid (e.g., 0.36°×0.36°) cannot fully represent them. The elongated distribution characteristics of aircraft emissions indicate that evenly allocating emissions around the airport will cause significant uncertainty. Due to variations in the MLH, the height at which aircraft emit pollutants during LTO can reach up

to 4544 m above the maximum altitude of 915 m set by the ICAO. The NOx emissions above the 915 m vary by month, accounting for an average of 24.6% (9.9%–37.5%) in the LTO cycle.

Our 4D-LTO emission inventory dataset reflects the actual spatial and temporal and can be used to accurately assess the air quality impact of aircraft in the LTO cycle. This dataset and our methodology play a vital role in an in-depth study of temporal and spatial variations of aircraft emissions and their health and environmental impact. By conducting in-depth

analysis of our refined dataset, we can quantify the aviation industry's contribution to climate change and explore potential emission reduction pathways. Furthermore, by adjustments to accommodate regional differences, e.g., operational activity data, airport-specific emission factors, and airport-specific flight trajectory datasets, our methodology possesses broad applicability and flexibility. The application of our methodology to other regions, is a fundamental in formulating effective strategies and policies to achieve global aviation emission reduction targets.

## 6 Author contributions

JL: conceptualization, formal analysis, funding acquisition, writing (original draft). ZY: methodology, software, visualization, project administration, writing (original draft). YZ: methodology, writing (review and editing), supervision. CW: investigation, visualization. XC: visualization, investigation.

## 7 Competing interests

The author has declared that they and their co-authors have no competing interests.

## 8 Acknowledgements

We are indebted to the company (Variflight.com) for providing the research data. In addition, we greatly appreciated Beijing Municipal Commission of Education and Beijing Municipal Commission of Science and Technology for supporting this work. The authors are grateful to the anonymous reviewers for their insightful comments.

## 9 Financial supports

This research was supported by the National Natural Science Foundation of China (No. 91644110) and National Key Research and Development Program of China (No. 2018YFC0213206).

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
