# Peer review of "Four-dimensional aircraft emission inventory dataset of Landing and takeoff cycle in China (2019–2023)"

_Earth System Science Data, 2024_

## Author Comment (AC1)

**Anonymous Referee #1:**

**Comments to the Author**

**Comment 1:** The aircraft engine emissions have important impacts on air quality in and around airports and the potential exposure of nearby residential populations. The impact study of aircraft emission relies on the detailed and accurate emission information. This paper provides a detailed information about four-dimensional aircraft emission for landing and takeoff cycle from 2019 to 2023 based on the flight time and trajectory information. It could provide useful basis to further study of the environmental impacts. Overall, this MS is well-structured and is appropriate for the scope of the Earth System Science Data journal. There are several necessary revisions should be made before the manuscript could be considered for publication acceptance.

Response: We are very grateful to the referee for the insightful review. The comments have contributed much to improve the manuscript. According to the referee's suggestions, we have conducted a revision. Each comment has been addressed on a point-by-point basis, with the referee's comments are noted in black, the responses to the referees' comments are noted in blue, and the corresponding revisions in the main text are noted *in (black) italic fonts*. All the changes are also marked in Revised Manuscript. We hope that this revised version of the manuscript addresses all of the reviewer's concerns.

**Comment 2:** Introduction: It should be stated how much China aircraft emissions contributes to global aircraft emissions. This provides a general context for global implications in terms of pollution that emphasizes the importance of better estimates of the specific emissions mentioned in this study.

Response: We thank the referee for the advice. We have added the explanation of the contributes of aircraft emissions in China in line 77–81 in Section 1 of the revised manuscript:

*"As the world's second-largest aviation market (CAAC), China contributes 13% of*

*global flight operations (Graver et al., 2020), and accounting for 7.8% to 23.5% of global aviation-related pollutant and carbon emissions (Ma et al., 2024; Teoh et al., 2024). Improving the accuracy of aviation emission estimates and enhancing temporal-spatial resolution in China can not only promote the green development of the Chinese aviation industry but also exert a far-reaching impact of global aircraft pollution mitigation."*

**Comment 3:** What is the content of Section 2.1.2?

Response: We sincerely thank you for your careful check. The missing section may be caused by the typesetting and format conversion. We have added the corresponding content in lines 140–154 in P5–6:

*"The daily maximum mixing layer height (MLH) serves as a key parameter for determining climb and approach modes of flight operations, and varies with region and time. Given data accessibility constraints, we substituted daily maximum MLH with the daily maximum planetary boundary layer height (PBLH), which shares analogous dynamic characteristics. The three steps for calculating climb and approach times are as follows.*

1) *Different airport daily maximum PBLHs in 2019–2023 were obtained based on Weather Research and Forecasting (WRF) model. The model parameter settings are described in our previous study (Wen et al., 2023).*

2) *The relationship between flight time and height were established. In our previous study (Zhou et al., 2019), the relationship for different airports in different months under the approach and climb mode was built based on Aircraft Meteorological Data Relay (AMDAR) data. AMDAR includes the aircraft's position (longitude, latitude, and altitude), speed, and associated meteorological parameters which were collected by the aircraft navigation system. The recording intervals are set at 6 s for the first 60 s of the climb phase, followed by once every 35 s thereafter, and once every 60 s during the descent phase. The form of the relationship for climb and approach mode can be found in Text A1 of SI. The $R^2$ (p < 0.001) of the*

*functional relationships of the climb and approach mode were above 0.93.*

3) *Each flight's actual climb and approach times from 2019 to 2023 were calculated based on the relationship of climb and approach mode mentioned above, and the daily maximum PBLH at different airport."*

**Comment 4:** Section 2.1: The time-in-mode was described in detail in emissions calculation, however, other input data such as emission factors for different flight modes (taxi, takeoff, climb, and approach) were not sourced or calculated.

Response: We thank the referee for the advice. We have added the description of EI and FF calculation method in lines 126–134 in Section 2.1.1 of the revised manuscript:

*"The EI of an aircraft type in different modes were calculated as (2):*

$$EI_{i,m,j} = n_i \times \sum_k EI_{k,m,j} \times P_{i,k}, \qquad (2)$$

*where $EI_{i,m,j}$ is the emission index of aircraft type $i$ in mode $m$ (g/kg) of pollutant $j$ (NOx, HC, and CO); $n_i$ is the number of engines fitted to aircraft type $i$; $EI_{k,m,j}$ is the emission index of engine $k$ in mode $m$ of pollutant $j$ (g/kg); and $P_{i,k}$ is the proportion of aircraft type $i$ equipped with engine $k$.*

*The FF of an aircraft type in different modes were estimated as (3).*

$$FF_{i,m} = n_i \times \sum_k FF_{k,m} \times P_{i,k}, \qquad (3)$$

*where $FF_{i,m}$ is the fuel flow of aircraft type $i$ in mode $m$ (kg/s); $FF_{k,m}$ is the fuel flow of engine $k$ in mode $m$ (kg/s); and the definitions of other parameters are similar to those used in (3)."*

**Comment 5:** Provide a table summarizing the emission factors used for key pollutants or cite the references.

Response: We thank the referee for the advice. We have added the Table SX to present the calculation method of emission index for different pollutants. In addition, we have added the description in line 135–138 in Section 2.1.1 of the revised manuscript:

*"In addition, the first-order approximation 3.0 (FOA3.0) (Wayson et al., 2009) method was used to recalculate the EI of PM, which is not included in EEDB. The emission factor of $SO_2$ is related to the sulphur content of jet fuel, so we used 3.868 g/kg as the emission factor of $SO_2$ (GB6537). In summary, the reference of the EI for different pollutants were shown in Table S1."*

**Table S1:** The reference of the emission index (EI) for different pollutants.

| Pollutant | Reference |
|-----------|-----------|
| HC | EEDB |
| CO | EEDB |
| NOx | EEDB |
| PM | EEDB; Wayson et al., 2009 |
| $SO_2$ | GB6537 |

**Comment 6:** Section 2.2: It is recommended to cite more classical literature on the application of the DBSCAN algorithm in this field.

Response: We thank the referee for the advice. We have added more classical literature on the application of the DBSCAN algorithm, including the application in the identification of road traffic, ship, and aircraft trajectories.

We have added the description in line 212–216 in Section 2.2.2 of the revised manuscript:

*"DBSCAN is a density-based clustering algorithm widely used in machine learning and data mining (Chen et al., 2021; Tekin et al., 2024). For the transportation industry, it is used for the identification research of road traffic, ship, and aircraft trajectories (Gui et al., 2021; Deng et al., 2023; Li et al., 2023). The DBSCAN algorithm belongs to unsupervised learning, and the initial value setting does not significantly affect the clustering results (Ventorim et al., 2021)."*

**Comment 7:** While the DBSCAN algorithm is referenced for flight trajectory recognition, the paper does not provide a detailed explanation of its parameters (e.g., minimum points, radius).

Response: We thank the referee for the advice. The DBSCAN algorithm relies on two input parameters, the minimum number of samples (MinPts) and distance threshold ($\varepsilon$). MinPts determines the minimum number of points required to form a dense region, while $\varepsilon$ specifies the maximum distance between two points to be considered as within the same neighbourhood.

We have added the description in line 224–227 in Section 2.2.2 of the revised manuscript:

*"The DBSCAN algorithm relies on two input parameters, the minimum number of samples (MinPts) and distance threshold ($\varepsilon$), to cluster the data space based on three basic concepts: directly density-reachable, density-reachable, and density-connected (Sander et al., 1998). MinPts determines the minimum number of points required to form a dense region, while $\varepsilon$ specifies the maximum distance between two points to be considered as within the same neighbourhood…"*

**Comment 8:** Section 3: What is the basis for determining the high-resolution spatial grid ($0.03° \times 0.03° \times 34$ vertical layers)?

Response: We thank the referee for the advice. This high-resolution grid was chosen to accurately capture the horizontal and altitude distribution characteristics of aircraft emissions during the LTO cycle, while also enabling the integration of the emission inventory into numerical models for subsequent research.

We have added the description in line 286–302 in Section 2.3.2 of the revised manuscript:

*"For the horizontal resolution, most airport runways are approximately 3–4 kilometers (CAAC) in length and certain pollutants (such as CO) are predominantly emitted during taxiing, i.e., on the runway. $0.03°\times0.03°$ is capable of reflecting the horizontal*

*distribution characteristics of aircraft emission. In addition, 0.03° × 0.03° is also a common resolution for air quality models. Therefore, the horizontal resolution of the 4D-LTO emission inventory is 0.03° × 0.03° with the latitude and longitude range of 3.40°N–53.56°N and 73.44°E–135.09°E, respectively.*

*For the altitude resolution, while ICAO defines the LTO cycle with a fixed mixing layer height (915 m), in reality, the mixing layer height varies significantly with region and time, leading to variations in the altitude range of the LTO cycle. Therefore, to better reflect the vertical distribution of aircraft emissions above 915 m during the LTO cycle, this study set the altitude range from 0 m to 15668 m. In addition, to ensure that the emission inventory can be effectively used in air quality models, this study used the air quality model commonly used 35-layer sigma stratification strategy (Wolfe et al., 2016). Therefore, the altitude resolution was divided into 34 layers from 0 m to 15668 m (0.0 m–38.3 m, 38.3 m–76.7 m, 76.7 m–115.3 m, 115.3 m–154 m, 154 m–231.8 m, 231.8 m–310.3 m, 310.3 m–389.3 m, 389.3 m–469 m, 469 m–549.3 m, 549.3 m–630.3 m, 630.3 m–711.9 m, 711.9 m–794.2 m, 794.2 m–960.7 m, 960.7 m–1130.1 m, 1130.1 m–1302.3 m, 1302.3 m–1477.6 m, 1477.6 m–1656.0 m, 1656.0 m–1929.7 m, 1929.7 m–2211.1 m, 2211.1 m–2599.3 m, 2599.3 m–3107.2 m, 3107.2 m–3643.1 m, 3643.1 m–4210.5 m, 4210.5 m–4813.9 m, 4813.9 m–5458.5 m, 5458.5 m–6151.2 m, 6151.2 m–6900.4 m, 6900.4 m–7717.4 m, 7717.4 m–8617.3 m, 8617.3 m–9621.2 m, 9621.2 m–10759.7 m, 10759.7 m–12080.6 m, 12080.6 m–13664.8 m, 13664.8 m–15668 m.)."*

**Comment 9:** Does this resolution significantly improve the representation of emissions compared to conventional models with fewer layers?

Response: We thank the referee for the advice. While ICAO defines the LTO cycle with a fixed mixing layer height (915 m), in reality, the mixing layer height varies significantly with region and time, leading to variations in the altitude range of the LTO cycle. However, conventional models with fewer layers cannot reflect the vertical emission distribution characteristics above 915 m during the LTO cycle. Therefore, our resolution improves the representation of emissions above 915 m during the LTO cycle

We have added the description in line 291–294 in Section 2.3.2 of the revised manuscript:

*"For the altitude resolution, while ICAO defines the LTO cycle with a fixed mixing layer height (915 m), in reality, the mixing layer height varies significantly with region and time, leading to variations in the altitude range of the LTO cycle. Therefore, to better reflect the vertical distribution of aircraft emissions above 915 m during the LTO cycle, this study set the altitude range from 0 m to 15668 m."*

**Comment 10:** Section 3.2: The data in this paragraph is messy to show as a list, can it be shown as a table or some other form?

Response: We thank the referee for the advice. To present the data more clearly, we have added the Table S6 of the SI to present the change of the LTO number during and after the COVID-19 pandemic. In addition, we have added the description in line 376–378 in Section 3.2 of the revised manuscript:

*"As can be seen from Figure 5(a) and Table S5, from January 20 to February 13, 2020, aircraft activity rapidly dropped to the lowest point owing to the impact of COVID-19, showing that the number of LTO on February 13, 2020, was 84.8% lower than the same period in 2019."*

**Table S5:** The difference of LTO numbers before, during and after the epidemic compared with the same period in 2019

| Date | LTO number $(10^4)$ | LTO number for the same period in 2019 $(10^4)$ | LTO number change rate compared the same period in 2019 |
|---|---|---|---|
| 2020.1.20 | 1.71 | 1.56 | 9.2% |
| 2020.2.13 | 0.24 | 1.61 | -84.8% |
| 2021.2.12 | 0.28 | 1.65 | -83.1% |
| 2021.8.12 | 0.67 | 1.70 | -60.3% |
| 2021.11.9 | 0.63 | 1.55 | -59.5% |

| | | | |
|---|---|---|---|
| 2022.4.4 | 0.19 | 1.57 | -87.7% |
| 2022.11.29 | 0.30 | 1.53 | -80.4% |
| 2023.4.6 | 1.45 | 1.45 | -0.1% |

**Comment 11:** Section 4: The study area is China, and the literature of the comparative study is better supplemented with more studies of Chinese airports.

Response: We thank the referee for the valuable advice. We have added more literatures of Chinese airports in line 466–468 in Section 3.4 of the revised manuscript:

*"Figure 8 uses the NOx emissions in January 2023 to show the differences between the 4D-LTO emission inventory and the LTO emission divided in previous study (Mokalled et al., 2018; Bo et al., 2019; Lawal et al., 2022; Wang et al., 2023; Zhang et al., 2023) in terms of height distribution (Fig. 8(a)–(b)) and horizontal distribution (Fig. 8(c)–(e))…"*

**Comment 12:** For the result of emission during 2020-2023, while the authors compare results with previous ICAO-based methods, the statistical measures of validation (e.g., $R^2$, RMSE) are not clearly presented.

Response: We thank the referee for the advice. We have utilized the Mean Absolute Error (MAE) and Mean Absolute Percentage Error (MAPE) as statistical indicators to evaluate the discrepancies between our 4D-LTO emissions inventory dataset and the previous ICAO-based methods. We have added the description in line 469–470 in Section 3.4 of the revised manuscript:

*"Two statistical measures, Mean Absolute Error (MAE) and Mean Absolute Percentage Error (MAPE), were employed to quantify these differences…"*

**Comment 13:** It should be further clarified the datasets used for validation, including observational data from airports and other inventory results.

Response: We thank the referee for the valuable advice. In Section 2.1, this study used the actual data to verify the running time data (taxi in, taxi out, climb, approach) used in the calculation of emissions. In Section 2.2, this study used the actual flight trajectory data to verify the performance of the DBSCAN model. In addition, we also to compared our dataset with previous study's spatial allocation methods. The description of the comparison in line 307–313 in Section 2.4 of the revised manuscript:

*"Our dataset was compared with the spatial allocation methods commonly used in previous studies. (1) Other studies typically assign aircraft emissions in the LTO cycle according to the standard altitude for each mode as defined by ICAO (Mokalled et al., 2018; Bo et al., 2019; Wang et al., 2023; Zhang et al., 2023). (2) The conventional horizontal distribution method for aircraft emissions in the LTO cycle assumes that aircraft emissions are radially distributed (Lawal et al., 2022). The Federal Aviation Administration (FAA)-recommended the standard climb rate of 200 ft per nautical mile. Therefore, the standard climb rate and ICAO standard altitude determine the horizontal distribution of aircraft emissions around the airport. The running time, altitude, and horizontal range of each mode defined by ICAO are shown in Table 1…"*

**Comment 14:** Line 444: The results mention a rebound in emissions by 2023 to 95.3% of 2019 levels, but this observation is not broken down by pollutant or flight mode.

Response: We thank the referee for the advice. We have added the description of the rebound in emissions for different pollutants in line 544–545 in Section 5 of the revised manuscript:

*"However, in 2023, the emissions of different pollutants quickly bounced back to 82.9%–94.1% of the 2019 levels, resulting in HC, CO, NOx, PM, and $SO_2$ emissions of 3.2 Gg, 46.1 Gg, 62.3 Gg, 1.1 Gg, and 18.4 Gg, respectively."*

**Comment 15:** Apart from NOx, other pollutants such as HC and PM are not discussed in detail, why? and what are their specific temporal and spatial patterns?

Response: We thank the referee for the advice. In this study, the temporal variations of different pollutants were introduced in Section 3.2. Regarding the spatial patterns, NOx makes a significant contribution to overall aircraft emissions and has a substantial impact on air quality. Furthermore, the spatial distribution of PM and SO$_2$ emissions from aircraft is similar to that of NOx. Notably, HC and CO are predominantly emitted during the taxi mode (Yang et al., 2018), which occurs on the runway. During the LTO cycle, emissions of HC and CO at other altitudes are negligible. Consequently, HC and CO emissions are concentrated in the first layer of the grid where the runway is located. In summary, this study primarily focused on the spatial distribution of NOx emissions. We have added the description in line 399–403 in Section 3.3 of the revised manuscript: *"During the LTO cycle, HC and CO emissions, predominantly emitted during taxi mode (Yang et al., 2018). Consequently, HC and CO emissions are distributed in the first layer of the grid where the runway is located. NOx is an important contributor to overall aircraft emissions and has a significant impact on air quality (Zhang et al., 2024). Furthermore, the spatial distribution of PM and SO$_2$ emissions from aircraft is similar to that of NOx. In summary, this study mainly analyzed the spatial distribution of NOx emissions."*

**Comment 16:** Line 460: The emissions above 915 m account for 24.6%, what is the significance of this finding? Does this altitude range impact local air quality differently than ground-level emissions?

Response: We thank the referee for the advice. The ICAO defines the LTO cycle with a fixed mixing layer height of approximately 915 m. However, in reality, the mixing layer height varies significantly with region and time, resulting in variations in the altitude range of the LTO cycle. Previous studies (Köhler et al., 2008; Lee et al., 2013; Yim et al., 2015; Zhang et al,.2023) that presents the high-altitude (above 915 m) emissions can significantly impact ground-level air quality through atmospheric transport and chemical reactions. This study found that some of these high-altitude

emissions were belong to the LTO cycle, and this part of emissions accounted for 24.6% of the total emissions during the LTO cycle. Therefore, ignoring these emissions would bring uncertainty in subsequent assessments of the impact of the LTO cycle on air quality and health.

We have added the description of the significance of our finding in line 459–462 in P19 of the revised manuscript:

*"Based on previous study (Köhler et al., 2008; Lee et al., 2013; Yim et al., 2015; Zhang et al., 2023), high-altitude emissions can significantly impact ground-level air quality through atmospheric transport and chemical reactions. When assessing emissions during the LTO cycle and their impact on air quality and health, we must fully consider the contribution of above 915 m emissions."*

**Comment 17:** Health impacts are mentioned in the introduction, but there is no specific health-related discussion in the discussion, specific pollutants such as NOx and PM$_{2.5}$ are known to cause respiratory and cardiovascular issues, please add or cite references.

Response: We thank the referee for the valuable advice. We have added the health-related descriptions and references in line 346–349 in Section 3.1 of the revised manuscript:

*"Emissions of pollutants from aircraft, such as NOx and PM$_{2.5}$, are known to cause respiratory and cardiovascular issues (Boningari et al., 2016; Hu et al., 2022; Hou et al., 2024). Therefore, it is essential to pay attention to the growing trend of aircraft activities in order to anticipate and address its potential health impacts."*

**Comment 18:** The dataset is established for China, how can the methodology be applied to other regions with different aviation?

Response: We thank the referee for the advice. While our 4D-LTO emission inventory dataset was initially established for China, the methodology possesses broad applicability. When applying our methodology to other regions with different aviation

profiles, researchers can directly construct airport-specific models by adhering to the 4D emission inventory method detailed in Section 3.2, and inputting localized parameters e.g., operational activity data, airport-specific emission factors, and airport-specific flight trajectory datasets.

We have added the description in line 566–568 in Section 5 of the revised manuscript:

*"Furthermore, by adjustments to accommodate regional differences, e.g., operational activity data, airport-specific emission factors, and airport-specific flight trajectory datasets, our methodology possesses broad applicability and flexibility."*

**Comment 19:** Linking the conclusion to wider global challenges such as climate change or international emissions reduction targets for aviation will be better.

Response: We thank the referee for the valuable advice. We have added the description of the related to climate change and international emissions reduction targets for aviation in line 564–569 in Section 5 of the revised manuscript:

[revised manuscript text omitted]

---

## Author Comment (AC2)

**Anonymous Referee #2:**

**Comment 1:** This study presents a four-dimensional aircraft emission inventory of the LTO cycle in China. The author first calculated the air pollutant emissions during the LTO cycles with the ICAO method with modeled running time and aircraft type-specified emission factors. Then flight altitude and horizontal trajectory was identified for emission allocations. The developed methods aim to construct the flight trajectories without using the ADS-B data that have limited availability. This study objective is attractive; however, the describing of the methods lacks some critical details and the model performance is not fully evaluated. More information is needed for readers to evaluate the quality of the presented emission data.

**Response:** We are very grateful to the referee for the insightful review. The comments have contributed much to improve the manuscript. According to the referee's suggestions, we have conducted a more detailed description of the methods and the evaluation of the model performance. Each comment has been addressed on a point-by-point basis, with the referee's comments are noted in black, the responses to the referees' comments are noted in blue, and the corresponding revisions in the main text are noted in (black) italic fonts. All the changes are also marked in Revised Manuscript. We hope that this revised version of the manuscript addresses all of the referee's concerns.

**Comment 2:** To calculate the flight emission, the author collected all aircraft types and proportion of engine types for each aircraft type to get a weighted EI and EF. Thus the accuracy of the engine proportion significantly affect the calculation accuracy. Are the proportion data with complete coverage and varying year by year? If not, please provides the details and add uncertainty discussion.

**Response:** We greatly appreciate your valuable comments. According to our investigation (Airbus; Aircraft Commerce), most aircraft types do not update engine.

Some aircraft types (e. g., the aircraft A allow is allowed to be equipped with engine A and B) may change engine configuration proportion during 2019–2023. Given the unavailability of the annual variation of engine configuration for each aircraft type in the existing datasets, this study used the latest proportion data of each aircraft type in different years.

We have added the limitation of the proportion data in line 518–523 in P21 of the revised manuscript:

*"Our 4D-LTO emission inventory dataset reflects the actual spatial and temporal and can be used to accurately assess the air quality impact of aircraft in the LTO cycle, but has several limitations due to data and technical restrictions. (1) According to our investigation (Airbus; Aircraft Commerce), most aircraft types do not update engine. Some aircraft types (e. g., the aircraft A allow is allowed to be equipped with engine A and B) may change engine configuration proportion during 2019–2023. Given the unavailability of the annual variation of engine configuration for each aircraft type in the existing datasets, this study used the latest proportion data of each aircraft type in different years…"*

**Comment 3:** The section 2.1.2 Climb and approach time calculation is missing.

Response: Thank you so much for your careful check. The missing section may be caused by the typesetting and format conversion. We have added the corresponding content in lines 140–154 in P5–6:

*"The daily maximum mixing layer height (MLH) serves as a key parameter for determining climb and approach modes of flight operations, and varies with region and time. Given data accessibility constraints, we substituted daily maximum MLH with the daily maximum planetary boundary layer height (PBLH), which shares analogous dynamic characteristics. The three steps for calculating climb and approach times are as follows.*

*1) Different airport daily maximum PBLHs in 2019–2023 were obtained based on Weather Research and Forecasting (WRF) model. The model parameter settings*

*are described in our previous study (Wen et al., 2023).*

2) *The relationship between flight time and height were established. In our previous study (Zhou et al., 2019), the relationship for different airports in different months under the approach and climb mode was built based on Aircraft Meteorological Data Relay (AMDAR) data. AMDAR includes the aircraft's position (longitude, latitude, and altitude), speed, and associated meteorological parameters which were collected by the aircraft navigation system. The recording intervals are set at 6 s for the first 60 s of the climb phase, followed by once every 35 s thereafter, and once every 60 s during the descent phase. The form of the relationship for climb and approach mode can be found in Text A1 of SI. The $R^2$ (p < 0.001) of the functional relationships of the climb and approach mode were above 0.93.*

3) *Each flight's actual climb and approach times from 2019 to 2023 were calculated based on the relationship of climb and approach mode mentioned above, and the daily maximum PBLH at different airport."*

**Comment 4:** The author constructed a model to estimate the aircraft's taxi time in order to fill the missingness in actual taxi time data; however, there lacks figures or tables to summarize the coverage and quality of actual taxi time data that were used for model fitting, making it hard to evaluate the representativeness of the model. The consistency of exponential relationship between deltaT and $T_0$ with N in different years need to be presented by fitting the exponential relationship with each year's data separately and comparing the fitted parameter. The $R^2$ of deltaT and N is 0.1 for taxi in in PEK, indicating poor correlation. The fitting effect of deltaT and $T_0$ in other airports is nor presented. A summary of the overall performance of the model is critical for readers to consider this method.

Response: We thank the referee for the advice. For the coverage and quality of the used data, the actual taxi time data in this study is recorded by the VariFlight. It used the information from ADS-B system, which is recognized by researchers as a reliable data source (Klenner et al., 2022; Zhang et al., 2022; Teoh et al., 2024). Limited by random

incompleteness of aircraft equipped with ADS-B equipment, this study has collected all available taxi time data, with coverage rates ranging from 47.9% to 67.0% during 2019–2023, which were summarized in Table S2. It can represent the taxiing conditions of aircraft at different airports with different operating scales.

This study developed the exponential relationship between $\Delta T$ and $T_0$ with N in different years for the estimation of corresponding year. Taking PEK as an example, we have discussed consistency of exponential relationship between $\Delta T$ and $T_0$ with N in different years. The exponential relationship between $\Delta T$ and $T_0$ with N in different years were shown in Table S4. The proposed five-year exponential relationship in Figure 2 of the manuscript could be a reference for the other study with no fitting data. We also calculated the coefficient of variation (CV, 30.4% for taxi out and 10.4% for taxi in operations) between the $\Delta T$ and $T_0$ estimation result from five-year model and specific year model, at a representative flight number of 20 (common across all study years). Compared with the actual taxi time, the estimation error of the model result (11.4% for taxi in mode and 20.4% for taxi out mode) is lower than the result based on the fixed ICAO standard taxi time (27.8% for taxi in mode and 22.0% for taxi out mode). About the low $R^2$ value for taxi in at PEK, Table S4 also presents that the performance of exponential relationships between $\Delta T$ and $T_0$ with N in different years are better.

These models only are used in the situation that the $\Delta T$ and $T_0$ could not be counted due to the lack of the record. In addition, taking 2023 as an example, we have added the description of the overall performance of the taxi time calculate model for other airports. In summary, the description for the coverage and quality of actual taxi time data were added in line 156–166 in Section 2.1.3. The description for the consistency and effect of the relationship in different years, and the taxi time calculation model effect of other airports was added in line 174-192 in Section 2.1.3:

*"ICAO specifies the taxi mode's running time (taxi out 19 min; taxi in 7 min). However, the actual taxi time varies based on airport flight schedules during actual operation, and using a fixed time can lead to emissions calculation uncertainty. Therefore, the*

*actual taxi time was used to calculate the aircraft's taxi emissions accurately. The actual taxi time data were obtained from the VariFlight based on the information of ADS-B system, which is recognized by researchers as a reliable data source (Klenner et al., 2022; Zhang et al., 2022; Teoh et al., 2024).*

*Since not all aircraft record the actual taxi time and the actual taxi time is not publicly available, this study has collected all available taxi time data, with coverage rates ranging from 47.9% to 67.0% during 2019–2023, which were summarized in Table S2. It can represent the taxiing conditions of aircraft at different airports with different operating scales. The missing taxi time were supplemented based on the hourly-airport difference relationship model between taxi time and aircraft number of schedules. The functional relationship between the number of aircraft on schedule and the taxi time is as follows…"*

**Table S2:** The proportion of flights with recorded taxi time.

| Year | Taxi in mode | Taxi out mode |
|------|--------------|---------------|
| 2019 | 47.9% | 48.4% |
| 2020 | 57.2% | 57.8% |
| 2021 | 53.8% | 50.2% |
| 2022 | 48.6% | 51.5% |
| 2023 | 59.3% | 67.0% |

*"…The taxi time relationship construction method was used to update the database from 2020 to 2023. The performance of the taxi time calculating model for different airports were shown in Figure S1 and Table S3, taking 2023 as an example. In addition, in this study, Beijing Capital International Airport (PEK) is chosen as a case to test the performance of taxi time model (Figure 2(a) and Figure 2(b)) under diverse flight situation (e.g., high-density scenarios), due to the centralized terminal layout and relatively frequent ground congestion (Liu et al., 2024).Taking 12:00 from 2019 to 2023 for the Beijing Capital International Airport (PEK) as an example, Fig. 2(a), Fig. 2(b) represent the comparative verification of function relationships for taxi in and taxi out in different years. We observed a strong correlation between taxi time and the number*

*of scheduled aircraft, regardless of whether it is taxi in or taxi out. The significance level (p < 0.001) indicates a strong relationship. The $R^2$ for taxi out ranges from 0.87 to 0.98, and for the taxi in mode, it ranges from 0.96 to 0.99. The model has a good effect on taxi in or out mode at different years, indicating that the model reflects the real taxi time variation.*

*If the relationship between taxiing time and the number of aircraft scheduled cannot be fitted at a certain time due to lack of records, Table S4 presents the exponential relationship of $\Delta T$ and $T_0$ at different years. In addition, Fig. 2 also provided the five-year exponential relationship of $\Delta T$ and $T_0$, which could be a reference for the other study with no fitting data. We also calculated the coefficient of variation (CV, 30.4% for taxi out and 10.4% for taxi in operations) between the $\Delta T$ and $T_0$ estimation result from five-year model and specific year model, at a representative flight number of 20 (common across all study years). Compared with the actual taxi time, the estimation error of the model result (11.4% for taxi in mode and 20.4% for taxi out mode) is lower than the result based on the fixed ICAO standard taxi time (27.8% for taxi in mode and 22.0% for taxi out mode). $\Delta T$ and $T_0$ estimation models only are used in the situation that the $\Delta T$ and $T_0$ could not be counted due to the lack of the record."*

**Table S4:** The fitting parameters and performance between N and $\Delta T$ or $T_0$ at PEK in different years.

| Year | Taxi out | | | | Taxi in | | | |
| | $\Delta T$ | | $T_0$ | | $\Delta T$ | | $T_0$ | |
| | Relationship | $R^2$ | Relationship | $R^2$ | Relationship | $R^2$ | Relationship | $R^2$ |
|---|---|---|---|---|---|---|---|---|
| 2019 | $\Delta T = 66.07e^{-0.027}$ | 0.96 | $T_0 = 625.71e^{-0.011}$ | 0.59 | $\Delta T = 21.01e^{-0.017}$ | 0.69 | $T_0 = 418.49e^{-0.020}$ | 0.96 |
| 2020 | $\Delta T = 49.43e^{-0.043}$ | 0.62 | $T_0 = 710.86e^{-0.023}$ | 0.83 | $\Delta T = 16.69e^{-0.015}$ | 0.39 | $T_0 = 422.65e^{-0.022}$ | 0.82 |
| 2021 | $\Delta T = 55.63e^{-0.049}$ | 0.89 | $T_0 = 621.92e^{-0.031}$ | 0.84 | $\Delta T = 25.88e^{-0.029}$ | 0.31 | $T_0 = 363.97e^{-0.025}$ | 0.88 |
| 2022 | $\Delta T = 36.58e^{-0.060}$ | 0.55 | $T_0 = 576.79e^{-0.028}$ | 0.63 | $\Delta T = 14.73e^{-0.023}$ | 0.19 | $T_0 = 380.45e^{-0.022}$ | 0.94 |
| 2023 | $\Delta T = 61.27e^{-0.039}$ | 0.78 | $T_0 = 611.48e^{-0.020}$ | 0.80 | $\Delta T = 18.39e^{-0.018}$ | 0.50 | $T_0 = 417.40e^{-0.023}$ | 0.89 |

[Figure]

**Figure S1:** Hourly $R^2$ of taxi in and out time calculating model for different airports.

**Table S3:** $R^2$ of between N and $\Delta T$ or $T_0$ of taxi in and out mode for different airports.

| Airport type | Taxi out | | | | Taxi in | | | |
|---|---|---|---|---|---|---|---|---|
| | $\Delta T$ | | $T_0$ | | $\Delta T$ | | $T_0$ | |
| | Mean | Std. | Mean | Std. | Mean | Std. | Mean | Std. |
| 4F | 0.86 | 0.09 | 0.58 | 0.19 | 0.59 | 0.20 | 0.74 | 0.15 |
| 4E | 0.57 | 0.24 | 0.38 | 0.18 | 0.44 | 0.22 | 0.63 | 0.23 |
| 4D | 0.41 | 0.20 | 0.35 | 0.11 | 0.34 | 0.17 | 0.45 | 0.21 |
| 4C | 0.41 | 0.19 | 0.39 | 0.20 | 0.28 | 0.18 | 0.30 | 0.14 |

**Comment 5:** The flight altitude was modeled following a previous study. The author should summarize this model here rather than only provide a citation.

Response: Thanks for the above suggestion. In this study, the flight altitude at different times was identified using the models mentioned in Section 2.1.2. We have added the description in lines 201–206 in P8 of the revised manuscript, and added the model summary in Text S1 of the SI:

*"The relationship between flight height and time (Zhou et al., 2019), which has been introduced in Section 2.1.2, was used to identify the altitude at different times for each LTO cycle. The daily maximum PBLH was used to identify the maximum height of each LTO cycle at different airports. When the taxi in and out is 0 m, the takeoff is from 0 m to 152 m (ICAO). The climb is from 152 m to PBLH, and the approach is from PBLH*

to 0 m. By integrating the altitude information with the emission inventory data established in Section 2.1, we were able to further vertically stratify the pollutant emission inventory during the LTO cycle."

"The relationship between height and time could be described as linear or quadratic functions. The general form of the functional relationships can be expressed in (A1).

$$H = aT^2 + bT + c \ (a \geq 0, T \geq 0) \qquad\qquad (A1)$$

where $H$ is aircraft height (m); $T$ is duration of the aircraft traverse from ground to the given height (climb) or from that height to the ground (approach) (s); a, b and c are constants in the equation. For airports without AMDAR data, the functional relationship of the nearest airport was employed."

**Comment 6:** The AMDAR data were used to identify flight horizontal trajectory, but information on the AMDAR data is missing.

Response: We thank the referee for the advice on the information of the AMDAR. We have added the description of the AMDAR data in lines 147–151 of the revised manuscript:

"...based on Aircraft Meteorological Data Relay (AMDAR) data. AMDAR includes the aircraft's position (longitude, latitude, and altitude), speed, and associated meteorological parameters which were collected by the aircraft navigation system. The recording intervals are set at 6 s for the first 60 s of the climb phase, followed by once every 35 s thereafter, and once every 60 s during the descent phase."

**Comment 7:** The description of the application of DBSCAN method is not clear. What are the inputs of DBSCAN? Why the trajectory is high dimension (besides longitude, latitude, and altitude) and the dimension was set to 25?

Response: We thank the referee's advice about the description of the application of DBSCAN method. Each flight trajectory contains four dimensions: time, longitude, latitude and altitude. Among them, the flight altitude of each flight could be confirmed

based on the method mentioned in Section 2.2.1. Therefore, the input of the DBSCAN method was the horizontal position information (time, longitude and latitude) for each flight trajectory.

About the dimension of the flight trajectory, we apologize for the confusion to the referee. The 25 represents the number of sampling points set at the same time interval for each flight trajectory.

This study used a unified indicator for assessing the similarity of flight trajectories: if the departure or arrival flight trajectories at different times are approximately in the same position during the same time period after takeoff or before landing, then these flight trajectories are considered similar. Based on AMDAR data information description in Section 2.1.2, the recording intervals are set at 6 s for the first 60 s of the climb phase, followed by once every 35 s thereafter, and once every 60 s during the descent phase. However, due to variations in flight trajectories and potential recording delays, the time intervals in the flight trajectory sequences are not uniform. To address this issue, we have decided to adopt a resampling method. The same number of sampling points for both the departure and arrival phases were set to ensure sampling consistency and uniformity across these two phases. According to actual flight data from 2019–2023, during the LTO cycle, departure was within 480 s and arrival was within 1200 s. We comprehensively considered the recording intervals of AMDAR data, the time range of the LTO cycle, and computational complexity, and found three optional integer division sampling numbers including 21 (Sampling interval: 24 s of departure, 60 s of arrival), 25 (Sampling interval: 20 s of departure and 50 s of arrival) and 31 (Sampling interval: 18 s of departure and 40 s of arrival). During the arrival process, 31 sampling points would result in largest discrepancy (20 s) with the AMDAR record interval. During the departure process, 21 sampling points would result in largest discrepancy (18 s) with the AMDAR record interval before the 60 s; 31 sampling points would result in largest discrepancy (19 s) with the AMDAR record interval after the 60 s. In addition, there are most computationally intensive for the option 31. Therefore, this study selected 25 sampling points.

We have added the detailed description of the DBSCAN methods in line 212–237 in

*"DBSCAN is a density-based clustering algorithm widely used in machine learning and data mining (Chen et al., 2021; Tekin et al., 2024). For the transportation industry, it is specifically used for the identification research of road traffic, ship, and aircraft trajectories (Gui et al., 2021; Deng et al., 2023; Li et al., 2023). The DBSCAN algorithm belongs to unsupervised learning, and the initial value setting does not significantly affect the clustering results (Ventorim et al., 2021). As a result, the DBSCAN algorithm is well suited for flight trajectory clustering processing with unclear information, such as the number of clusters and distribution characteristics (Murça et al., 2018; Giovanni et al., 2024).*

*Before clustering, flight trajectory data belonging to the LTO cycle should be extracted from a vast amount of information in AMDAR. First, the climb and approach modes in the LTO cycle are screened according to the ascending and descending symbols in AMDAR information. Second, each flight trajectory was divided into airport ownership according to the airport's location. Finally, the horizontal position information (time, longitude and latitude) of each flight trajectory in the climb and approach modes of different airports was obtained as the input information for flight trajectory clustering. The DBSCAN algorithm relies on two input parameters, the minimum number of samples (MinPts) and distance threshold ($\varepsilon$), to cluster the data space based on three basic concepts: directly density-reachable, density-reachable, and density-connected (Sander et al., 1998). MinPts determines the minimum number of points required to form a dense region, while $\varepsilon$ specifies the maximum distance between two points to be considered as within the same neighbourhood.*

*DBSCAN is good at calculating the distance between points, but it is difficult for DBSCAN to process the flight trajectory with time attribute in this study (Chen et al., 2021). Therefore, we use the Euclidean norm to compute the distance between the two sets of flight trajectories. The premise of using the Euclidean norm is to keep the time interval of each set of flight trajectories the same. However, the time interval of each flight trajectory sequence is not the same because of each flight's trajectory difference and recording delay. As a result, we conducted unified processing of each departure*

*and arrival trajectory through the resampling method. Too low and too high of sampling points will make the location feature information unclear and increase the computational complexity of clustering processing, respectively. Based on all actual flight data from 2019–2023, during the LTO cycle, departure was within 480 s and arrival was within 1200 s. To comprehensively considered the recording intervals of AMDAR data, the uniformity across departure and arrival phases, and computational complexity, we set the sampling points of each trajectory to 25.”*

**Comment 8:** The performance of taxi time model is represented by PEK and the performance of trajectory cluster is represented by PVG. Why these two airports are selected? I am wondering the model performance at other airports.

Response: We thank the referee's advice. PEK airport is chosen as a case to test the performance of taxi time model under diverse flight situation (e.g., high-density scenarios), due to the centralized terminal layout and relatively frequent ground congestion (Liu et al., 2024). PVG airport is chosen as a case to test the performance of trajectory cluster under the normal flight trajectories scenario, as well as the deviations in flight trajectories due to the crosswinds and typhoons which is the challenge for the robustness of the trajectory cluster algorithm (Wang et al., 2017; Xu et al., 2020).
According to the comment, we have added the description to demonstrate performance of the taxi time model and trajectory clustering for different airports. The reason of the PEK selection and the description of the taxi time model performance at other airports is presented in line 174–178 in Section 2.1.3. The reason of the PVG selection and the description of the trajectory cluster performance at other airports is presented in line 245–248 in Section 2.2.2:
*"The performance of the taxi time calculating model for different airports were shown in Figure S1 and Table S3, taking 2023 as an example. In addition, in this study, Beijing Capital International Airport (PEK) is chosen as a case to test the performance of taxi time model (Figure 2(a) and Figure 2(b)) under diverse flight situation (e.g., high-density scenarios), due to the centralized terminal layout and relatively frequent ground*

*congestion (Liu et al., 2024)."*

[Figure]

**Figure S1:** Hourly $R^2$ of taxi in and out time calculating model for different airports.

**Table S3:** $R^2$ of between N and $\Delta T$ or $T_0$ of taxi in and out mode for different airports.

| Airport type | Taxi out | | | | Taxi in | | | |
| --- | --- | --- | --- | --- | --- | --- | --- | --- |
| | $\Delta T$ | | $T_0$ | | $\Delta T$ | | $T_0$ | |
| | Mean | Std. | Mean | Std. | Mean | Std. | Mean | Std. |
| 4F | 0.86 | 0.09 | 0.58 | 0.19 | 0.59 | 0.20 | 0.74 | 0.15 |
| 4E | 0.57 | 0.24 | 0.38 | 0.18 | 0.44 | 0.22 | 0.63 | 0.23 |
| 4D | 0.41 | 0.20 | 0.35 | 0.11 | 0.34 | 0.17 | 0.45 | 0.21 |
| 4C | 0.41 | 0.19 | 0.39 | 0.20 | 0.28 | 0.18 | 0.30 | 0.14 |

*"Figure S2 shows the overall performance of the trajectory clustering model for different airports. In addition, in this study, Shanghai Pudong International Airport (PVG) is chosen as a case to test the performance of trajectory cluster (Figure 3) under the normal flight trajectories scenario, as well as the deviations in flight trajectories due to the crosswinds and typhoons which is the challenge for the robustness of the trajectory cluster algorithm (Wang et al., 2017; Xu et al., 2020)."*

[Figure]

**Figure S2**: The R and MAE for flight trajectories at different airports.

**Comment 9:** The uncertainty calculation section is full of uncertainty. How did the author calculate the distribution parameters of each terms with uncertainty and what are these parameters used in the final calculation? The uncertainty ranges were only presented at two airports with PEK shows uncertainty in emission calculation and PVG shows uncertainty in spatial distribution. What is the overall uncertainty of this emission inventory?

Response: We thank the referee's advice about the uncertainty section. For the calculate method of the distribution parameters of each term, this study comprehensively considered the uncertainty of EI, FF and T when calculating the uncertainty of emissions. Furthermore, this study used the Monte Carlo sampling method to obtain the uncertainty of the emission for different pollutants with 20,000 samples.

In this study, the spatial uncertainty during the LTO cycle including the uncertainty of horizontal and altitude position. The standard deviation of the horizontal position is calculated by the error distribution between the flight trajectory clustering result and the actual flight trajectory. The standard deviation of the altitude position is the combine

of the standard deviation of a function fitting parameter a, b, and c. This study employed the Monte Carlo method to quantitatively assess the uncertainty of spatial location identification for each hour, with uncertainty ranges derived from 20,000 Monte Carlo simulations at a 95% prediction interval.

The description of overall uncertainty of the emission inventory has been added to replace the discussion of typical airports.

In order to make the uncertainty method description more clearly, we added the new section (Section 2.5, Line 317–338 in P12) to introduce the calculation method in detail:

*"2.5 Uncertainty calculation*

*The uncertainty of the 4D-LTO emission inventory dataset including emission calculation uncertainty and spatial location identification uncertainty. This study assumes that the uncertainty of all input parameters follows a normal distribution.*

*When calculating the emissions uncertainty, this study comprehensively considered the uncertainty of EI, FF and T. The EI and FF are weighted based on the engines data from the EEDB and the engine proportion data for different aircraft types. Therefore, the standard deviation of EI or FF were calculated using the formula (7):*

$$\sigma = \sqrt{\sum_{k}(x_k - \bar{x})^2 \times P_k}, \tag{7}$$

*where $\sigma$ represents the standard deviation of EI or FF, $k$ represents the engine type, $x_k$ represents the EI or FF value of the engine k, $\bar{x}$ represents the weighted average of EF or FF, and $P_k$ represents the proportion of engine k.*

*The climb and approach time is obtained using the relationship between flight time and flight height (Zhou et al.,2019). Therefore, the standard deviation of the climb and approach time is the combine of the standard deviation of a function fitting parameter a, b, and c. The taxi in/out time is calculated using the formula (4). Therefore, the standard deviation of the taxi in and out time is the combine of the standard deviation of a function fitting parameter $\Delta T$ and $T_0$. This study used the Monte Carlo sampling method to obtain the 95% prediction interval of the emission for different pollutants with 20,000 samples.*

*The spatial uncertainty during the LTO cycle include the uncertainty of horizontal and*

*altitude position. The standard deviation of the horizontal position is calculated by the error distribution between the flight trajectory clustering result and the actual flight trajectory. The standard deviation of the altitude position is the combine of the standard deviation of a function fitting parameter a, b, and c. This study employed the Monte Carlo method to quantitatively assess the uncertainty of spatial location identification for each hour, with uncertainty ranges derived from 20,000 Monte Carlo simulations at a 95% prediction interval."*

The Section 3.5 (Line 496–505 in P21) has presented the statistical results of the uncertainty for emissions and spatial location:

*"Taking the year 2023 as an example, this study estimated the hourly uncertainty ranges for different emissions in various airports throughout the year, with the average of the uncertainty ranges for NOx, CO, HC, PM, and $SO_2$ being [-17%, 17%], [-16%, 16%], [-6%, 6%], [-8%, 8%], and [-8%, 8%] respectively, and their standard deviations being (±11%), (±10%), (±3%), (±4%), and (±4%) respectively.*

*This study also estimated the hourly average 95% prediction intervals for latitude and longitude of departure and arrival flights at various airports, with the average of [-0.02°, 0.02°] (longitude) and [-0.02°, 0.02°] (latitude) for departure, and [-0.06°, 0.06°] (longitude) and [-0.05°, 0.05°] (latitude) for arrival, respectively, and their standard deviations being (±0.01°), (±0.01°), (±0.01°), and (±0.01°) respectively. In addition, the result showed that the hourly 95% prediction intervals of climb and approach altitude at different airports within an average of [-78 m, 78 m] and [-134 m, 134 m], respectively, and their standard deviations being (±49 m) and (±95 m), respectively."*

**Comment 10:** Some previous studies, e.g. Teoh et al., 2024 and Zhang et al., 2022, used ADS-B data to estimate the aviation emissions at a high spatiotemporal resolution. What is the major advantage of this study compared to those previous studies? Please also compare the estimated aviation emissions to other emission inventories, e.g. EDGAR, EMEP, AERO2k et al.

[revised manuscript text omitted]